# Sampling and Identity-Testing Without Approximate Tensorization of Entropy

**William Gay** [1]   **William He** [2]   **Nicholas Kocurek** [3]   **Ryan O'Donnell** [2]

## Abstract

We study the problems of approximate sampling from and distribution testing of *mixture models*, where the modes satisfy a functional inequality called *approximate tensorization of entropy* (ATE). While it is known that ATE makes these tasks more efficient in the unimodal setting, mixtures of few distributions satisfying ATE do not necessarily satisfy ATE overall, leading to a lack of theoretical guarantees for multimodal distributions, which are a key challenging case of modern generative models. We show this gap can be overcome by establishing the following pair of results for mixtures of ATE distributions:

1. We show fast mixing of Glauber dynamics from a *data-based initialization*, with *optimal* sample complexity, for mixtures of distributions satisfying modified log-Sobolev inequalities, building on similar results in (Koehler & Vuong, 2024; Huang et al., 2024) for mixtures satisfying the weaker Poincaré inequality.

2. Answering an open question from (Blanca et al., 2023), we give efficient identity-testers for mixtures of ATE distributions in the coordinate-conditional sampling access model.

## 1. Introduction

Markov Chain Monte Carlo methods, where one samples from a distribution of interest by running a Markov chain which converges to said target, are ubiquitous in both the practical and theoretical study of machine learning, statistics,

[1]Univerisity of Illinois Urbana-Champaign, Champaign, IL, USA [2]Carnegie Mellon University, Pittsburgh, PA, USA [3]University of Washington, Seattle, WA, USA. Correspondence to: William He <wrhe@cs.cmu.edu>.

*Proceedings of the 43rd International Conference on Machine Learning*, Seoul, South Korea. PMLR 306, 2026. Copyright 2026 by the author(s).

and computational physics. Often theoretical interest lies in bounding the mixing time, how long it takes for a chain to converge given it starts in the worst possible configuration. In understanding valid criteria for fast mixing, we have also developed a characterization of what makes a Markov chain fail to mix, and it may come as no surprise to the empirical machine learning community that the culprit is prevalently multimodality (Sly & Sun, 2012; Galanis et al., 2015). Even distributions as simple as a mixture of two mean-separated Gaussians or sufficiently distant discrete product measures can provably make Langevin or Glauber dynamics chains fail to mix.

Despite these established worst-case results, many settings where MCMC is utilized do not require such a stringent condition as mixing from an arbitrary configuration (Hinton, 2002; Xie et al., 2016; Nijkamp et al., 2020). In the setting of generative modeling, for example, one attempts to sample from a distribution given a small set of training data to learn from. This application has led to an increase of interest in *beyond worst-case mixing*, establishing guarantees for Markov chains under additional structural assumptions (Koehler & Vuong, 2024; Koehler et al., 2024; Huang et al., 2024). Our primary assumption, based off the setting of generative modeling, is the introduction of a *data-based initialization*, an empirical distribution based on a small number of samples from the true distribution. Our main contribution utilizes such an initialization in the sampling and identity-testing of multimodal distributions where the modes satisfy a functional-analytic property called approximate tensorization of entropy.

### 1.1. Approximate Tensorization of Entropy

Let $\mu$ be a distribution on the discrete product set $\Sigma^n$. If $f$ is a (non-negative) real-valued function on $\Sigma^n$, then the following functional captures the amount of local variation that $f$ has, where "local" means with respect to varying a single component of $\Sigma^n$:

**Definition 1.** We write $\mathcal{L}_\mu$ for the functional on functions $f : \Sigma^n \to \mathbb{R}_{\geq 0}$ given by

$$\mathcal{L}_\mu[f] = \sum_{i \in [n]} \mathop{\mathbf{E}}_{\boldsymbol{x} \sim \mu} \left[ \mathop{\mathbf{Ent}}_{\boldsymbol{y} \sim \mu|_{\boldsymbol{x}_{\setminus i}}} [f(\boldsymbol{y})] \right].$$

Here $\mathbf{Ent}_\mu[\cdot]$ is the standard entropy functional with respect to $\mu$. See Section 2 for a full definition.

Every product distribution $\mu$ on $\Sigma^n$ satisfies *tensorization of entropy*, meaning $\mathcal{L}_\mu[f] \geq \mathbf{Ent}_\mu[f]$ for all $f$. Much work in the context of Markov chain mixing is focused on establishing an approximate version of this inequality for distributions of interest:

**Definition 2.** A distribution $\mu$ on $\Sigma^n$ satisfies *approximate tensorization of entropy (ATE)* with constant $c^*$ if for all $f : \Sigma^n \to \mathbb{R}_{\geq 0}$,

$$\mathbf{Ent}_\mu[f] \leq c^* \cdot \mathcal{L}_\mu[f].$$

Note that $c^* \geq 1$ unless $\mu$ is a point mass (in which case both sides are always 0).

A motivation for Definition 2 is the fact that many statistical tasks related to $\mu$ become easier when $\mu$ satisfies $c^*$-ATE with a small $c^*$ (close to 1).

**Sampling.** One such statistical task is that of approximately sampling from $\mu$, given oracle access to $\mu$ up to proportionality. When $\mu$ satisfies $c^*$-ATE, then it is well-known that the Glauber dynamics Markov chain for $\mu$ mixes in time $\widetilde{O}(c^* n)$, and Markov chain Monte Carlo techniques are very effective for drawing approximate samples from $\mu$. Thus, establishing ATE is instrumental for obtaining optimal mixing times for Glauber dynamic chains for natural distributions on high-dimensional spaces, such as Gibbs distributions of certain spin systems at high temperatures. See, for example, (Bobkov & Tetali, 2006; Caputo et al., 2015; Anari et al., 2021; Chen et al., 2021b; Chen & Eldan, 2022; Blanca et al., 2022; Hermon & Salez, 2023; Chen et al., 2023; 2024).

**Identity-Testing.** Another statistical task where ATE helps significantly is identity-testing, also known as hypothesis-testing or goodness-of-fit testing. In this setting, a testing algorithm is given some distance measure $D$ between distributions, a threshold parameter $\varepsilon$, access to some kind of description of a known ("visible") distribution $\mu$, and some kind of sample access to an unknown distribution $\pi$. The tester must then satisfy the following performance guarantees:

1. If $\pi = \mu$ then the tester accepts $\pi$ with high probability.

2. If $D(\pi \parallel \mu) \geq \varepsilon$ then the tester rejects $\pi$ with high probability.

The study of the complexity of this problem has a long history in statistics (Pearson, 1900; Fisher, 1966). This is a fundamental problem in science, where one wants to confirm that the behavior of some system conforms to that of a purported model for that system.

Motivated by previous works on identity-testing with alternative access models (see Section 1.4 for a more detailed account of the literature), Blanca, Chen, Stefankovic, and Vigoda (Blanca et al., 2023) studied the model of coordinate conditional access, which is a common relaxation of subcube conditioning access and pairwise conditional access when $\Sigma$ is binary. In this access model, the tester gets access to samples from the unknown distribution $\pi$, and also gets access to $\pi|_{x_{\setminus i}}$ for any $x \in \Sigma^n$. Here $x_{\setminus i} = \{y \in \Sigma^n : |x - y|_0 \leq 1\}$. Blanca et al. proved that distributions $\mu$ satisfying ATE admit very efficient identity testers.

More precisely, let the "General Oracle" be an oracle that, when queried, outputs a sample drawn from the unknown distribution $\pi$, and let the "Coordinate Oracle" be the oracle that, when queried with a pair $(x, i)$, outputs a sample from $\pi|_{x_{\setminus i}}$.

**Theorem 3** ((Blanca et al., 2023), Theorem 4.1). *Let $\mu$ be a distribution on $\Sigma^n$ that is $\eta$-balanced (see Definition 30), fully supported, and ATE with constant $c^*$. Assuming*

$$c^* \leq \mathrm{poly}(n), \quad \eta \geq \exp(-\mathrm{poly}(n)),$$

*there is a testing algorithm for $\mu$ with access to the Coordinate Oracle and General Oracle having*

$$\text{sample complexity} \leq O\left(\frac{c^* n}{\varepsilon}\right) \cdot \log^3\left(\frac{n}{\varepsilon}\right) \cdot f(\eta),$$

*where* $f(\eta) = \begin{cases} \log(1/\eta) & \text{if } |\Sigma| = 2, \\ \frac{1}{\sqrt{\eta}} & \text{if } |\Sigma| \geq 3. \end{cases}$

Also, in the $|\Sigma| \geq 3$ case one can reduce the dependence on $1/\eta$ back to logarithmic at the expense of being quadratic in the other parameters; precisely, one can also achieve

$$\text{sample complexity} \leq O\left(\frac{c^* n}{\varepsilon}\right)^2 \cdot \log^2\left(\frac{n}{\varepsilon}\right) \cdot \sqrt{|\Sigma|} \cdot \log(1/\eta).$$

### 1.2. Mixtures of ATE Distributions and Our Results

As described above, certain statistical tasks become much easier when the related distribution $\mu$ satisfies approximate tensorization of entropy, which leads us to consider cases in which this property fails. A very natural class of distributions that do not satisfy ATE is the class of mixtures of distributions that *do* satisfy ATE. We consider distributions $\mu$ of the form $\sum_{a \in [k]} \rho(a)\mu_a$, where $\rho$ is a distribution on $[k]$ and each $\mu_a$ satisfies $c^*$-ATE. We call the $\mu_a$'s the *mixture components*. Such multimodal distributions arise naturally in many machine learning contexts.

Our first result concerns the sampling task. It is easy to see that certain mixtures of ATE distributions have exponentially large mixing times for Glauber dynamics (for example the equal mixture of the 0.1- and 0.9-biased distributions on $\{0, 1\}^n$). However, this is a lower bound on the mixing time from a *worst-case initialization*. We instead show that when the chain is initialized with a *data-based initialization*, it still experiences fast mixing, given that there is enough data for the initialization:

**Theorem 4.** Let $\mu = \sum_{a=1}^k \rho(a)\mu_a$ be a mixture on $\Sigma^n$ with each component satisfying $c^*$-ATE (or, more weakly, satisfying $\frac{1}{c^*n}$-MLSI; see Definition 15 and Remark 17). Let $\boldsymbol{\pi}$ be an empirical distribution induced by

$$m = O(k/\varepsilon + \log(1/\delta)/\varepsilon) \tag{1}$$

independent samples from $\mu$. Then with probability at least $1 - \delta$ over these samples, the Glauber dynamics for $\mu$ warm-started at $\boldsymbol{\pi}$ mixes to $D_{\mathrm{KL}}(P_t\boldsymbol{\pi} \parallel \mu) \leq \varepsilon$ in continuous-time $t = c^*n \cdot O(\log\log(1/\min_x \mu(x)) + \log(1/\varepsilon))$.

Since learning a distribution on $k$ elements to $\varepsilon$ KL-divergence requires $\Theta(k/\varepsilon + \log(1/\delta)/\varepsilon)$ (Canonne, 2020), this result is tight in terms of sample complexity. Additionally, while we only prove Theorem 4 in the context of discrete mixtures over a product space $\Sigma^n$, our results should hold for any finite component mixture of distributions satisfying a modified log-Sobolev inequality, granted we modify the Markov chain accordingly (e.g. Langevin dynamics for continuous domains).

Our next result concerns the identity-testing task. We show that with the same access model studied by (Blanca et al., 2023), we can efficiently identity-test any $\mu$ that is a mixture of few ATE distributions, answering the open question posed in that paper:

**Theorem 5.** Let $\mu = \sum_{a=1}^k \rho(a)\mu_a$ be a mixture on $\Sigma^n$ with each component satisfying $c^*$-ATE. Assume also that $\mu$ is $\eta$-balanced (see Definition 30). Then there is an identity-testing algorithm for $\mu$ that uses

$$O\left(\frac{c^*n}{\varepsilon}\right) \cdot \log^2\left(\frac{c^*n}{\varepsilon}\right) \cdot \log(1/\eta) \cdot \log\log(1/\eta) \tag{2}$$

$$\cdot \sqrt{|\Sigma|} + O\left(\frac{\sqrt{k} \cdot \log(1/\rho^*)}{\varepsilon}\right) \tag{3}$$

calls to the General Oracle and Coordinate Oracle.

**Remark 6.** Our algorithm, and the algorithm of (Blanca et al., 2023), do not use the full power of the Coordinate Oracle. The algorithms even work in the setting where the set of calls that the tester can make to the Coordinate Oracle is predetermined as a random set of pairs $\boldsymbol{x} \sim \pi$ and $\boldsymbol{i} \in [n]$. Moreover, given an efficiently accessible description of the distribution $\mu$, the test has similarly efficient

computational complexity. As mentioned in (Blanca et al., 2023), this efficient description is equivalent to the ability to efficiently implement single steps of Glauber dynamics for the distribution $\mu$.

We defer the proof of Theorem 5 to Appendix A but give a proof overview at the end of the main body.

As mentioned before, in the case where $k = 1$, Theorems 4 and 5 are essentially known from previous work. In analyzing MCMC algorithms for sampling from $\mu$, one wants to show that the KL-divergence between the state of the Markov chain and $\mu$ is contracting. When analyzing identity-testing algorithms for $\mu$ one wants show that if an unknown distribution $\pi$ has large KL-divergence from $\mu$, then this is captured by the divergences between neighbors in $\Sigma^n$. The ATE inequality then relates the contraction of KL-divergence through the Markov chain and the divergences between neighbors to the original KL-divergence.

However, when $\mu$ is merely a mixture of distributions that individually satisfy ATE, such a local-to-global property fails. That is, the variation between $\mu$ and some alternative distribution $\pi$ is not conserved when zooming in from the entire distribution $\mu$ to $\mu|_{x \setminus i}$. In fact, when the variation between $\mu$ and $\pi$ is well-captured by the local variation, we are essentially done.

In order to deal with distributions $\pi$ whose variations are not captured locally, one needs to identify where exactly the variation *is* captured, and then deal with such cases accordingly. The chain rule for entropy (Lemma 18) allows us to identify where the variation is captured, and we develop new techniques to deal with such cases.

### 1.3. Related Work in Sampling

The task of sampling from multimodal distributions given a data-based initialization is natural in the context of generative modeling, in which the task is exactly to generate given a small batch of ground truth samples to model the underlying distribution. The term "data-based initialization" was originally coined in (Nijkamp et al., 2020) in the context of image generative modeling and the success of energy-based models with short-run Langevin trained unsupervised on a small sample. The success of such initializations can be seen in related frameworks such as contrastive divergence and other approximations for Maximum Likelihood Estimation (Hinton, 2002; Xie et al., 2016; Gao et al., 2018).

Our results fit in a line of work providing a theoretical framework and guarantees for the empirical success novel generative modeling techniques. Addressing the failure of score matching in multimodal sampling settings (Koehler et al., 2023), the work (Koehler & Vuong, 2024) showed introducing a data-based initialization can correct otherwise divergent trajectories in Langevin diffusion in the case of

a mixture of strongly log-concave distributions. This was followed up by a pair of works giving guarantees similar to our MCMC-based convergence (Glauber dynamics) for discrete mixtures satisfying a Poincaré inequality. Our result Theorem 4 can then be seen as an general entropic analogue of the works of Koehler, Lee, and Vuong (Koehler et al., 2024) and Huang, Mohanty, Rajaraman, and Wu (Huang et al., 2024). We then use this general entropic version to yield optimal results in the case of a modified log-Sobolev inequality, which follows from ATE.

More concretely, suppose each mixture component in $\mu$ satisfies a Poincaré inequality. Using higher-order spectral gaps for the Glauber dynamics chain, (Koehler et al., 2024) were able to establish that reasonably fast mixing in TV distance occurs from a warm-start having a near-minimal number of samples. However, due to their use of Poincaré inequalities, the mixing time they conclude can be suboptimal. They also prove a version of their main theorem with a faster mixing time using standard log-Sobolev inequalities, but this relies on the hypothesis that each mixture component satisfies a log-Sobolev inequality, which is a stronger assumption than an MLSI.

(Huang et al., 2024) obtains a result similar to (Koehler et al., 2024), again under the assumption that each mixture component satisfies a Poincaré inequality. However, their result is quantitatively much weaker in sample complexity: (1) it has a larger dependence on $k$ and $\varepsilon$; and, (2) it has a dependence on the minimum mixing weight $\rho_a$, which does not appear in (Koehler et al., 2024). (Huang et al., 2024) also suffers from suboptimal mixing time due to use of Poincaré inequalities, but it does achieve mixing in $\chi^2$-divergence, which is stronger than mixing in TV distance.

Our Theorem 4 simultaneously achieves: (1) optimal mixing times in TV distance in the case that $\mu$ is a product distribution, by using MLSIs rather than Poincaré inequalities; and, (2) optimal sample complexity in the case where $\mu$ is a mixture of isolated point masses. A key point in our analysis is that we avoid using standard Chernoff bounds, which seem to necessarily introduce a dependence on the minimum mixing weight, and instead employ a bound on the moment generating function (m.g.f.) for the KL-divergence of an empirical distribution from the true distribution.

We summarize these results in the table below.

| | Assumption | Sample Complexity |
|---|---|---|
| Koehler et al. | Poincaré | $O(k \log(k/\delta)/\varepsilon^2)$ |
| Huang et al. | Poincaré | $O(\log(1/\delta)/\min_a \rho_a \varepsilon^2)$ |
| This work | MLSI | $O(k/\varepsilon^2 + \log(1/\delta)/\varepsilon^2)$ |

*Table 1.* Comparison to previous work, showing the mixture assumption and sample complexity needed to guarantee $1 - \delta$ probability of $\varepsilon$-mixing in TV distance.

## 1.4. Related Work in Testing

Our Theorem 5 fits into a broader line of work on testing distributions with alternative sampling models, which is motivated by the fact that testing algorithms with access to i.i.d. samples from a high-dimensional distribution often require an exponential number of samples.

In the direction of providing more powerful models of access to distributions in identity-testing, concurrent works of Canonne, Ron, and Servedio (Canonne et al., 2014) and Chakraborty, Fischer, Goldhirsch, and Matsliah (Chakraborty et al., 2013) introduced the problem of identity-testing with access to conditional samples from $\pi$. That is, instead of getting access to i.i.d. samples from the unknown distribution $\pi$, the tester can specify a subset $S$ of the underlying set of outcomes and receive samples from $\pi|_S$. These papers showed that *any* distribution admits an identity-tester in this model whose sample complexity is $\text{poly}(1/\varepsilon)$, where $\varepsilon$ denotes the minimum distance of distributions that should be rejected with high probability. Subsequent work (Falahatgar et al., 2015) continued the study of testers with general conditional samples.

Other access models to the unknown distribution $\pi$ have also been considered. For example, (Canonne & Rubinfeld, 2014) considered the dual query and cumulative dual query models in which one can explicitly query the probability density function on points and subsets of the universe, respectively. See also later works (Caferov et al., 2015; Narayanan & Tetek, 2023).

A model in which one can sample from $\pi$ conditioned on an arbitrary subset of the universe is rather strong; in many settings it might be unclear how to simulate/obtain such samples. There are, however, natural weakenings of this access model in the high-dimensional setting. In particular, suppose $\Sigma^n$ is the set of configurations of a system of $n$ particles (or individuals, organisms, etc.). Then it might be significantly more reasonable to obtain samples from "subcubes" of $\Sigma^n$, meaning conditional distributions in which a subset of the particles have fixed states. Indeed, this subcube conditioning model was introduced by Bhattacharyya and Chakraborty (Bhattacharyya & Chakraborty, 2018), who showed that $\widetilde{O}(n^2)$ subcube-conditioned samples suffice for identity-testing of arbitrary distributions on $\Sigma^n$. Subsequent works (Canonne et al., 2021; Chen et al., 2021a) provided improvements in cases where assumptions on the visible distribution $\mu$ are made.

Another realistic relaxation of arbitrary conditioning studied was that of pairwise conditioning, or conditioning under subsets of size 2, which was also studied in (Canonne et al., 2014). Narayanan (Narayanan, 2021) provided testing algorithms for arbitrary distributions with complexity $\widetilde{O}\left(\frac{\sqrt{n}}{\varepsilon^2}\right)$.

We provide some further motivation for the coordinate conditional sampling access model not mentioned in (Blanca et al., 2023). Our motivation is based on practical matters, and we suggest a situation in which one might be able to easily simulate coordinate conditional access in cases where subcube conditional access and pairwise conditional access might not be easily simulable. In our situation, we again regard $\Sigma^n$ as a configuration of $n$ particles, and we think of $\pi$ as some distribution on the set of configurations of those $n$ particles, potentially the Gibbs distribution for some set of interactions between the particles.

A natural model for the evolution of the configuration over time is that of Glauber dynamics. In the Glauber dynamics process, given a current configuration $x \in \Sigma^n$, the next configuration is chosen from $\pi|_{x \setminus i}$. Assume the ability to i) arbitrary fix configurations of particles; and ii) simulate Glauber dynamics for a distribution $\pi$. Then to simulate access to $\pi|_{x \setminus i}$, one can repeatedly initialize the system to $x$ and simulate one step of Glauber dynamics for $\pi$ starting from $x$. Any step that updates the $j$th particle for $j \neq i$ is ignored, and the result given by an update to the $i$th particle is a sample from $\pi|_{x \setminus i}$. Note here that we assume that we can tell when a site undergoes resampling, even if the resampling does not result in a new state.

## 2. The Chain Rule of Entropy

In this section, we discuss $\Phi$-entropy and how its contraction can be used to bound the mixing time of Markov chains, and then show how the standard chain rule of entropy (often seen as the law of total variance) characterizes the mixing time of Glauber dynamics for mixture models.

### 2.1. Entropies and Divergences

The following notion of $\Phi$-entropy was introduced in (Chafaï, 2004):

**Definition 7** ($\Phi$-entropy). Let $\Phi$ be a smooth and convex function mapping some interval of real numbers to the nonnegative real numbers. Let $\mu$ be a probability distribution on a finite set $\Omega$. The $\Phi$-*entropy* of a function $f : \Omega \to \mathbb{R}$ with respect to $\mu$ is defined to be

$$\mathbf{Ent}_\mu^\Phi[f] := \mathop{\mathbf{E}}_{\boldsymbol{x} \sim \mu} [\Phi(f(\boldsymbol{x}))] - \Phi\left(\mathop{\mathbf{E}}_{\boldsymbol{x} \sim \mu}[f(\boldsymbol{x})]\right).$$

**Fact 8.** If $\Phi$ is convex then $\mathbf{Ent}_\mu^\Phi[f] \geq 0$.

**Remark 9.** Assume $\Phi(1) = 0$ and let $\pi$ and $\mu$ be probability distributions on $\Omega$. Then the $\Phi$-entropy functional

$$\mathbf{Ent}_\mu^\Phi\left[\frac{\pi}{\mu}\right] = D_\Phi(\pi \parallel \mu)$$

is also known as the $\Phi$-divergence[1] between $\pi$ and $\mu$. For

example,

$$\mathbf{Ent}_\mu^{u \mapsto u \log u}\left[\frac{\pi}{\mu}\right] = D_{\mathrm{KL}}(\pi \parallel \mu).$$

As usual, we use the convention $0 \log 0 = 0$, and that this function is defined on $\mathbb{R}_{\geq 0}$. The quantity $D_{\mathrm{KL}}(\pi \parallel \mu)$ may be $\infty$ (when $\pi \not\ll \mu$).

We are most often interested in the case $\Phi(u) = u \log u$ throughout the paper, so when we drop the $\Phi$ in the superscript, the case $\mathbf{Ent}_\mu = \mathbf{Ent}_\mu^{u \log u}$ is assumed. An important property of this specific choice of $\Phi$, used in our identity-testing result, is that the resulting entropy functional is 1-homogeneous:

**Fact 10.** $\mathbf{Ent}$ is 1-homogeneous: That is, if $\alpha$ is a nonnegative scalar, then $\mathbf{Ent}_\mu[\alpha f] = \alpha \mathbf{Ent}_\mu[f]$.

While we think of $D_\Phi$ as a measure of divergence between $\pi$ and $\mu$, it is not a proper metric, and thus it is usually preferred to study the total variation distance as a metric between distributions.

**Definition 11** (Total variation distance). Let $\mu, \pi$ be distributions on a discrete space $\Omega$. The total variation distance is the metric given by

$$d_{\mathrm{TV}}(\mu, \pi) = \frac{1}{2} \sum_{x \in \Omega} |\mu(x) - \pi(x)|.$$

### 2.2. Markov Chains

Let $P \in \mathbb{R}^{\Omega \times \Omega}$ be the transition matrix of a discrete-time aperiodic and irreducible Markov chain over discrete $\Omega$. The mixing time of $P$ is defined as follows.

**Definition 12** (Mixing time). Let $P$ have stationary distribution $\mu$.

$$t_{\mathrm{mix}}(P) = \max_{x \in \Omega} \min\{t \geq 0 \mid d_{\mathrm{TV}}(P^t \delta_x, \mu) < 1/4\}.$$

The fundamental quantity useful in the functional analysis of Markov chain mixing is the Dirichlet form.

**Definition 13** (Dirichlet form). Let $P$ have stationary distribution $\mu$. Then the Dirichlet form with respect to $P$ is defined on functions $f, g : \Omega \to \mathbb{R}$ as

$$\mathcal{E}_P(f, g) = \mathbf{E}_{\boldsymbol{x} \sim \mu} \mathbf{E}_{\boldsymbol{y} \sim P\boldsymbol{x}} [(f(\boldsymbol{x}) - f(\boldsymbol{y}))(g(\boldsymbol{x}) - g(\boldsymbol{y}))].$$

We study inequalities relating the Dirichlet form, which is a measure of local variation, to measures of global variation in the case where the associated Markov chain is the Glauber dynamics chain with respect to a distribution $\mu$ on $\Sigma^n$. We define the Glauber dynamics chain formally here.

---

[1] The notation of "$f$-divergence" is more common, but to avoid notational overload we use the symbol "$\Phi$."

**Definition 14** (Glauber dynamics). Let $\mu$ be a distribution over $\Sigma^n$. The Glauber dynamics chain for $\mu$ is the chain in which given a current state $x \in \Sigma^n$ the next state is sampled by sampling a uniform random $\boldsymbol{i} \in [n]$ and then sampling the next state $\boldsymbol{y} \sim \mu|_{x \setminus i}$. That is, any point $x \in \Sigma^n$ transitions to any $y \in \Sigma^n$ with $|x - y| = 1$ with probability

$$\mathbf{Pr}\left[x \to_P y\right] = \frac{1}{n} \frac{\mu(y)}{\mu(x) + \mu(y)},$$

and otherwise stays at $x$. Informally, the chain resamples single coordinates at a time in a way that ensures that $\mu$ is stationary.

To show bounds on the mixing time, one typically establishes a $\Phi$-Sobolev inequality relating the $\Phi$-entropy to a certain Dirichlet form dependent on $\Phi$.

**Definition 15** ($\Phi$-Sobolev inequality). A distribution $\mu$ on $\Sigma^n$ satisfies a $\Phi$-*Sobolev inequality* with constant $c^*$ if for all $f : \Sigma^n \to \mathbb{R}_{\geq 0}$,

$$\mathbf{Ent}_{\mu}^{\Phi}[f] \leq c^* \cdot \mathcal{E}_P(f, \Phi'(f)). \tag{4}$$

Here $P$ is the Glauber dynamics chain associated to $\mu$.

The importance of this notion comes from the fact that the right-hand side of Equation (4) governs the decay of the $\Phi$-entropy between a Markov chain's current distribution and the stationary distribution through time, which can then be used to deduce TV distance bounds.

**Remark 16.** Let $\Phi(x) = x \log x$ and notice that $\Phi' = x \mapsto 1 + \log x$. Since $\mathcal{E}_P$ is translation-invariant, we have $\mathcal{E}_P(f, \Phi'(f)) = \mathcal{E}_P(f, \log f)$, and therefore Equation (4) is the usual *modified log-Sobolev inequality*. Similarly, when $\Phi(x) = x^2$ we have $\mathcal{E}_P(f, \Phi'(f)) = 4\mathcal{E}(f, f)$, and Equation (4) is (up to factor 4) the *Poincaré inequality*.

**Remark 17.** It is known that $c^*$-ATE implies $\frac{2}{c^* n}$-MLSI. See Proposition 1.1 in (Caputo et al., 2015).

## 2.3. The Chain Rule

When $\Phi(u) = u^2$, the following Lemma 18 is known as the law of total variance. When $\Phi(u) = u \log u$, it is known as the chain rule for entropy. Both of these tools are of great use in establishing Poincaré and log-Sobolev inequalities. See, for example, (Lee & Yau, 1998; Salez, 2021).

The fact itself is standard (see, e.g., (Beigi & Gohari, 2018)), but we give a proof in Appendix B for the convenience of the reader:

**Lemma 18.** If $\mu = \sum_{a=1}^{k} \rho(a)\mu_a$, then $\mathbf{Ent}_{\mu}^{\Phi}[f] = \mathbf{Ent}_{\boldsymbol{a} \sim \rho}^{\Phi}\left[\mathbf{E}_{\mu_a}[f]\right] + \mathbf{E}_{\boldsymbol{a} \sim \rho}\left[\mathbf{Ent}_{\mu_a}^{\Phi}[f]\right]$.

Lemma 18 is especially useful when a distribution $\mu$ is a mixture of many distributions. In the case where each of

the mixture components satisfies approximate tensorization of entropy, we can use Lemma 18 to show that the local entropy of some function $f$ under the distribution $\mu$ is lower-bounded by the portion of the entropy of $f$ that arises as "intra-component" entropy.

**Lemma 19.** If $\mu = \sum_{a=1}^{k} \rho(a)\mu_a$, where each $\mu_a$ satisfies $c^*$-ATE, then for any distribution $\pi$ on $\Sigma^n$ we have

$$c^* \cdot \mathcal{L}_{\mu}[f] \geq \underset{\boldsymbol{a} \sim \rho}{\mathbf{E}}\left[\underset{\boldsymbol{x} \sim \mu_a}{\mathbf{Ent}}[f(\boldsymbol{x})]\right].$$

Lemma 19 is proved in Appendix B. Lemma 19 essentially shows that the intra-component contribution to the entropy of $f$ is captured by the local entropy in the case where the mixture components satisfy ATE. In the context of our sampling result, this implies that any initialization that is evenly balanced across the components will experience fast mixing. In the context of our identity-testing result, this implies that any unknown distribution that is evenly balanced across the components yet still far from the target distribution will be rejected by a local tester.

The main contribution of this paper is showing how to also handle the inter-component contribution in our applications. That is, we need to show that data-based initializations are evenly balanced with high probability and that our identity-testing algorithm will reject unbalanced distributions.

For this purpose it will be helpful to characterize the inter-component entropy $\mathbf{Ent}_{\boldsymbol{a} \sim \rho}[\mathbf{E}_{\mu_a}[f]]$ in the case that $f = \pi/\mu$ is a density function. We can notice that $g$ defined by $a \mapsto \mathbf{E}_{\mu_a}[f]$ is itself a density on $[k]$ vis-a-vis $\rho$. More explicitly, define $\rho_\pi$ to be the probability distribution on $[k]$ induced by sampling $\boldsymbol{x} \sim \pi$, and then drawing $\boldsymbol{a} \sim \rho_{\boldsymbol{x}}$ where $\rho_{\boldsymbol{x}}$ is the posterior of $\boldsymbol{x}$ from $\mu$ with respect to $\rho$. In other words

$$\rho_\pi(a) = \sum_{x \in \Omega} \pi(x) \cdot \frac{\rho(a)\mu_a(x)}{\mu(x)}.$$

Now we can observe that $g$ is indeed the density $\rho_\pi/\rho$; hence:

**Fact 20.** For any mixture $\mu = \sum_{a=1}^{k} \rho(a)\mu_a$ we have $\mathbf{Ent}_{\boldsymbol{a} \sim \rho}^{\Phi}[\mathbf{E}_{\mu_a}[\pi/\mu]] = D_\Phi(\rho_\pi \parallel \rho)$.

## 3. Sampling from Data-Based Initializations

In this section we prove Theorem 4. An important observation made in (Huang et al., 2024) is that the Glauber dynamics chain for a mixture of distributions satisfying either Poincaré or modified log-Sobolev inequalities satisfies a *weak Poincaré* or *weak modified log-Sobolev inequality* respectively. They then use these weak functional inequalities to infer fast mixing. We first generalize this result to the

setting of $\Phi$-Sobolev inequalities, through the following $\Phi$-entropic generalization of Theorem 4.5 from (Huang et al., 2024). See Appendix B for the proof.

**Lemma 21.** Let $\mu = \sum_{a=1}^k \rho(a)\mu_a$ be a mixture of distributions on $\Sigma^n$ with each $\mu_a$ satisfying a $\Phi$-Sobolev inequality with constant $c^*$. Let $P$ be the transition matrix for the Glauber dynamics for $\mu$ and $P_t$ be the associated continuous-time Markov operator. Then, for any initial distribution $\pi$ we have

$$D_\Phi(P_t\pi \parallel \mu) \tag{5}$$

$$\leq \left(1 - \frac{1}{c^*n}\right)^t D_\Phi(\pi \parallel \mu) + \mathbf{E}_{s}\left[\mathbf{Ent}^\Phi_{\boldsymbol{a}\sim\rho}\left[\mathbf{E}_{P_s\pi}\left[\frac{\mu_{\boldsymbol{a}}}{\mu}\right]\right]\right], \tag{6}$$

where $\boldsymbol{s}$ is some random variable supported on $[0, t]$.

Lemma 21 shows that as long as one initializes Glauber dynamics for $\mu$ at a distribution $\pi$ so that the second term on the right-hand side in Equation (5) is small, the chain will experience fast mixing at the rate of the individual mixtures. The question is then how to choose $\pi$ such that this quantity is indeed small. We show that when $\pi$ is a "data-based initialization", meaning the empirical distribution formed by some number of i.i.d. samples from $\mu$, the inter-component entropy is indeed small by concentration.

Recall Fact 20, which motivates us to study the inter-component entropy as the $\Phi$-divergence between the empirical posterior distribution $\rho_{\boldsymbol{\pi}} = \frac{1}{m}\sum_{j=1}^m \rho_{\boldsymbol{x}_j}$ and the mixing weights. If the mixture components are separated, the task becomes to learn a distribution on $[k]$ from samples up to $\varepsilon$ error in $\Phi$-divergence. Furthermore, by exactly characterizing the m.g.f. of the empirical estimator's KL-divergence, we can use a convexity argument to handle the case when components are not separated. While we focus here on the well-studied case of KL-divergence, to establish a version of Theorem 4 for any $\Phi$-divergence one generally only needs to establish the sample complexity of the learning task for a reasonable estimator (see (Canonne, 2020) for more on such learning tasks).

### 3.1. The Case of KL-Divergence

Our main result, Theorem 4, is to apply this paradigm in the case of KL-divergence. When all $\rho_{\boldsymbol{x}_j}$ are indicator vectors (corresponding to the case of disjointly supported mixture components) the result then directly follows by taking $m = \Theta(k/\varepsilon + \log(1/\delta)/\varepsilon)$ in the recent result of Agrawal given here:

**Theorem 22** ((Agrawal, 2020), Theorem I.2). For $\varepsilon > \frac{k-1}{m}$ we have that

$$\mathbf{Pr}_{\boldsymbol{a}\sim\rho}\left[D_{\mathrm{KL}}(\operatorname*{avg}_{j\in[m]} \delta_{\boldsymbol{a}_j} \parallel \rho) > \varepsilon\right] \leq e^{-\varepsilon m} \cdot \left(\frac{e\varepsilon m}{k-1}\right)^{k-1}.$$

This result is established immediately by the following m.g.f. bound when $\lambda = m - \frac{k-1}{\varepsilon}$.

**Theorem 23** ((Agrawal, 2020), Theorem I.3). For $0 \leq \lambda < m$ we have that

$$\mathbf{E}_{\boldsymbol{a}\sim\rho}\left[\exp\left(\lambda \cdot D_{\mathrm{KL}}(\operatorname*{avg}_{j\in[m]} \delta_{\boldsymbol{a}_j} \parallel \rho)\right)\right] \leq \left(\frac{1}{1-\lambda/m}\right)^{k-1}.$$

We apply a simple convexity argument to arrive at the same m.g.f. bound for the general case of overlapping mixture components:

**Lemma 24.** Let $\boldsymbol{\pi} = \frac{1}{m}\sum_{j=1}^m \delta_{\boldsymbol{x}_j}$, where each $\boldsymbol{x}_j$ is sampled i.i.d. from $\mu = \sum_{a=1}^k \rho(a)\mu_a$. Then we have the following moment generating function bound:

$$\mathbf{E}_{\boldsymbol{x}_1,\ldots,\boldsymbol{x}_m}[\exp(\lambda \cdot D_{\mathrm{KL}}(\rho_{\boldsymbol{\pi}} \parallel \rho))] \leq \left(\frac{1}{1-\lambda/m}\right)^{k-1}.$$

*Proof.* For all $\lambda > 0$ we have

$$\mathbf{E}_{\boldsymbol{x}_1,\ldots,\boldsymbol{x}_m}[\exp(\lambda \cdot D_{\mathrm{KL}}(\rho_{\boldsymbol{\pi}} \parallel \rho))]$$

$$= \mathbf{E}_{\boldsymbol{x}_1,\ldots,\boldsymbol{x}_m}\left[\exp\left(\lambda \cdot D_{\mathrm{KL}}(\operatorname*{avg}_{j\in[m]} \rho_{\boldsymbol{x}_j} \parallel \rho)\right)\right]$$

$$= \mathbf{E}_{\boldsymbol{x}_1,\ldots,\boldsymbol{x}_m}\left[\exp\left(\lambda \cdot D_{\mathrm{KL}}(\operatorname*{\mathbf{E}}_{\boldsymbol{a}_j\sim\rho_{\boldsymbol{x}_j}} \operatorname*{avg}_{j\in[m]} \delta_{\boldsymbol{a}_j} \parallel \rho)\right)\right]$$

$$\leq \mathbf{E}_{\boldsymbol{x}_1,\ldots,\boldsymbol{x}_m} \mathbf{E}_{\boldsymbol{a}_j\sim\rho_{\boldsymbol{x}_j}}\left[\exp\left(\lambda \cdot D_{\mathrm{KL}}(\operatorname*{avg}_{j\in[m]} \delta_{\boldsymbol{a}_j} \parallel \rho)\right)\right]$$

$$= \mathbf{E}_{\boldsymbol{a}_1,\ldots,\boldsymbol{a}_m\sim\rho}\left[\exp\left(\lambda \cdot D_{\mathrm{KL}}(\operatorname*{avg}_{j\in[m]} \delta_{\boldsymbol{a}_j} \parallel \rho)\right)\right].$$

The inequality follows from the convexity of the KL-divergence in the left component and the exponential function when $\lambda > 0$. Applying Theorem 23 completes the proof. $\square$

We can transfer this m.g.f. bound (and accordingly the tail bound) to that of a convex combination of these quantities over the $\boldsymbol{s}$:

**Lemma 25.** Let $\boldsymbol{\pi} = \frac{1}{m}\sum_{j=1}^m \delta_{\boldsymbol{x}_j}$, where the $\boldsymbol{x}_j$ are sampled i.i.d. from $\mu = \sum_{a=1}^k \rho(a)\mu_a$. Then we have the following tail bound for any $\boldsymbol{s}$ whenever $\varepsilon > \frac{k-1}{m}$:

$$\mathbf{Pr}_{\boldsymbol{x}_1,\ldots,\boldsymbol{x}_m}\left[\mathbf{E}_{\boldsymbol{s}}[D_{\mathrm{KL}}(\rho_{P_{\boldsymbol{s}}\boldsymbol{\pi}} \parallel \rho) > \varepsilon]\right]$$

$$\leq e^{-\varepsilon m} \cdot \left(\frac{e\varepsilon m}{k-1}\right)^{k-1}.$$

*Proof.* In general, let $\boldsymbol{X} = \mathbf{E}_{\boldsymbol{s}}[\boldsymbol{X}_{\boldsymbol{s}}]$ where each $\boldsymbol{X}_i$ is distributed identically. Fixing $\lambda \geq 0$, we have:

$$\mathbf{E}\left[e^{\lambda \boldsymbol{X}}\right] = \mathbf{E}\left[e^{\lambda \mathbf{E}_{\boldsymbol{s}}[\boldsymbol{X}_{\boldsymbol{s}}]}\right] \leq \mathbf{E}\left[\underset{\boldsymbol{s}}{\mathbf{E}}\left[e^{\lambda \boldsymbol{X}_{\boldsymbol{s}}}\right]\right]$$
$$= \underset{\boldsymbol{s}}{\mathbf{E}}\left[\mathbf{E}\left[e^{\lambda \boldsymbol{X}_{\boldsymbol{s}}}\right]\right] = \mathbf{E}\left[e^{\lambda \boldsymbol{X}_0}\right].$$

The inequality follows via convexity, and $\boldsymbol{X}_0$ is a copy of $\boldsymbol{X}_{\boldsymbol{s}}$.

Set $\boldsymbol{X}_{\boldsymbol{s}} = D_{\mathrm{KL}}(\rho_{P_s \boldsymbol{\pi}} \parallel \rho) = D_{\mathrm{KL}}(\mathrm{avg}_{j \in [m]} \rho_{P_s \delta_{\boldsymbol{x}_j}} \parallel \rho)$ for all $s$. Note that each $\boldsymbol{X}_{\boldsymbol{s}}$ is marginally distributed according to the distribution given by $D_{\mathrm{KL}}(\mathrm{avg}_{j \in [m]} \rho_{\delta_{\boldsymbol{x}_j}} \parallel \rho)$ since each $\boldsymbol{x}_j \sim \mu$, and $\mu$ is stationary with respect to $P$. Whenever $0 \leq \lambda < m$ we apply the above and get the bound

$$\mathbf{E}\left[e^{\lambda \mathbf{E}_{\boldsymbol{s}}[D_{\mathrm{KL}}(\rho_{P_{\boldsymbol{s}} \boldsymbol{\pi}} \parallel \rho)]}\right]$$
$$\leq \mathbf{E}\left[e^{\lambda D_{\mathrm{KL}}(\rho_{\boldsymbol{\pi}} \parallel \rho)}\right] \underset{\text{Lemma 24}}{\leq} \left(\frac{1}{1 - \lambda/m}\right)^{k-1}.$$

Taking $\lambda = m - \frac{k-1}{\varepsilon}$ as in Theorem 22 finishes the proof. $\square$

Combining Lemma 21 and Lemma 25 immediately implies this formal version of Theorem 4:

**Theorem 26.** Let $\mu = \sum_{a=1}^{k} \rho(a) \mu_a$ be a mixture distribution with each component satisfying a modified log-Sobolev inequality with constant $c^*$. Let $P_t$ be the continuous-time Markov operator induced by the Glauber dynamics for $\mu$ and $\boldsymbol{\pi} = \frac{1}{m}\sum_{j=1}^{m} \delta_{\boldsymbol{x}_j}$ for $\boldsymbol{x}_1, ..., \boldsymbol{x}_m \sim \mu$. Then

$$\mathbf{Pr}_{\boldsymbol{x}_1, ..., \boldsymbol{x}_m}\left[D_{\mathrm{KL}}(P_t \boldsymbol{\pi} \parallel \mu) > \varepsilon\right] \leq \delta,$$

for $m = \Theta(k/\varepsilon + \log(1/\delta)/\varepsilon)$ and $t = c^* n \cdot \Theta\left(\log\log(1/\min_x \mu_x) + \log(1/\varepsilon)\right)$.

*Proof.* By Lemma 21, the fact that $D_{\mathrm{KL}}(\pi \parallel \mu) \leq \log(1/\min_x \mu(x))$, and the setting of $t$ it suffices to show that for the claimed sample complexity $m$ we have

$$\mathbf{Pr}_{\boldsymbol{x}_1, ..., \boldsymbol{x}_m}\left[\underset{\boldsymbol{s}}{\mathbf{E}}\left[D_{\mathrm{KL}}(\rho_{P_{\boldsymbol{s}} \boldsymbol{\pi}} \parallel \rho)\right] > 0.5\varepsilon\right] \leq \delta.$$

The use of the correct constant in the setting of $m$ and Lemma 24 imply the desired bound. $\square$

## Identity-Testing with Coordinate Conditional Sampling Proof Overview

Our Algorithm 1 is more or less the same as the algorithm of (Blanca et al., 2023), except for Step 2 (and the use of an improved KL identity-tester). To motivate the design of our algorithm, consider a distribution $\pi$ that is far from $\mu$ in KL-divergence. Then, by the chain rule (Lemma 18), we have

that either $\mathbf{E}_{\boldsymbol{a} \sim \rho} \mathbf{Ent}_{\mu_a}\left[\frac{\pi}{\mu}\right]$ is large, or $\mathbf{Ent}_{\boldsymbol{a} \sim \rho} \mathbf{E}_{\mu_a}\left[\frac{\pi}{\mu}\right]$ is large. Intuitively, either the intra-component entropy is large, or the inter-component entropy is large.

In the first case, the algorithm of (Blanca et al., 2023) rejects $\pi$ with high probability. This does not directly follow from the guarantee of (Blanca et al., 2023), but another application of the chain rule allows us to deduce this. See Appendix A.1.1 for the analysis of this part of the algorithm..

In the second case, the inter-component entropy is large. However, it may be the case that the intra-component entropy is small. For example, $\pi$ could be a mixture of the $\mu_a$, but with incorrect mixture weights. In this case, it is not clear how to use the coordinate oracle to detect this discrepancy in a generic way, especially if there is some intra-component entropy. However, in this case, we note that we can use posterior sampling of $\boldsymbol{a} \sim \rho|\boldsymbol{x}$, where $\boldsymbol{x}$ are samples from $\pi$, to infer what the effective weights on each component are. Here $\rho|x$ is the distribution on $[k]$ where $(\rho|x)(a) = \frac{\rho(a)\mu_a(x)}{\mu(x)}$.

## Impact Statement

This paper presents work whose goal is to advance the field of Machine Learning. There are many potential societal consequences of our work, none which we feel must be specifically highlighted here.

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

# A. Identity-Testing with Coordinate Conditional Sampling

Our second application deals with the problem of identity-testing distributions on $\Sigma^n$, but with access to coordinate conditional samples from the target distribution $\mu$, which is the mixture of some number of distributions sampling ATE. We prove Theorem 5.

Continuing formally from our proof overview, the inter-component entropy is equal to the KL-divergence between this posterior distribution when $x \sim \pi$ and when $x \sim \mu$. Given knowledge of the true distribution $\rho$, we can again use our base KL-divergence tester from Theorem 27 to reject this $\pi$. This is the content of Step 1 in Algorithm 1, analyzed in Appendix A.1.2.

As a subroutine, we will need an identity-testing algorithm with respect to KL-divergence error $\varepsilon$, for $\eta$-balanced distributions on domains of size $d$ (we will use this algorithm for the cases $d = k$ and $d = |\Sigma|$). In (Blanca et al., 2023), two such testers were given and analyzed; a main one with sample complexity $O\left(\min\left\{\frac{\sqrt{d}\ln(1/\eta)}{\varepsilon^2}, \frac{1}{\sqrt{\eta}\varepsilon}\right\}\right)$,[2] and one for the special case of $k = 2$ with sample complexity $O\left(\frac{\ln(1/\eta)}{\varepsilon}\right)$. The main effort there was to get $\log(1/\eta)$ dependence, rather than $1/\eta$ dependence; however, the general case suffers from a quadratic dependence on $1/\varepsilon$.

**Theorem 27.** Let $q$ be a distribution over a universe $D$ of size $d$, with each outcome in $D$ having probability at least $\eta$ under $q$. There is an algorithm KL-TEST$(p, q, \varepsilon, \delta)$ that given input $\varepsilon, \delta > 0$ and access to samples from an unknown distribution $p$ on $D$, draws

$$O\left(\frac{\sqrt{d} \cdot \log(1/\eta) \cdot \log(1/\delta)}{\varepsilon}\right)$$

samples from $p$ and has the following performance guarantee:

1. If $p = q$ then KL-TEST$(p, q, \varepsilon, \delta)$ accepts with probability at least $1 - \delta$.

2. If $D_{\mathrm{KL}}(p \parallel q) \geq \varepsilon$ then KL-TEST$(p, q, \varepsilon, \delta)$ accepts with probability at most $\delta$.

Here we note a tester with both linear dependence on $1/\varepsilon$ and logarithmic dependence on $1/\eta$ can be recovered by combining two known results from the literature; we prove this in Appendix B:

**Remark 28.** The (Blanca et al., 2023) work shows that in Theorem 27, we can actually take the minimum of the sample complexity with $O\left(\frac{1}{\sqrt{\eta}\varepsilon}\right)$. For simplicity of exposition, we will not carry around this 'min' in our subsequent complexity bounds (for the reasons described in Footnote 2).

With this tester in hand, we may state our algorithm.

---

[2]We remark that the first expression in the min essentially always dominates. Even when its $\varepsilon$ dependence is made linear, as in our Theorem 27, the second expression is smaller only in the regime $\frac{1}{d\log^2 d} \lesssim \eta \leq \frac{1}{d}$, and then only by a $\log d$ factor.

---

**Algorithm 1** Identity-testing of $\mu$ with Coordinate Oracle and General Oracle access.

---

**Input:** Coordinate Oracle and General Oracle access (See Remark 6) to distribution $\pi$ on $\Sigma^n$, known mixing weights $\rho(1), \ldots, \rho(k)$, descriptions of $c^*$-ATE distributions $\mu_1, \ldots, \mu_k$ on $\Sigma^n$, and the assumption that distribution $\mu = \sum_{a=1}^{k} \rho(a)\mu_a$ is $\eta$-balanced.
**Output:** "Accept" or "Reject".

COORDINATE-ORACLE-TEST$(\pi, \mu, \varepsilon)$:

1. Independently draw $T_1 = O\left(\frac{c^*n}{\varepsilon}\right)$ pairs $(\boldsymbol{x}, \boldsymbol{i})$, where $\boldsymbol{x} \sim \pi$ and $\boldsymbol{i} \in [n]$ is uniformly random. For each pair $(\boldsymbol{x}, \boldsymbol{i})$, reject if KL-TEST$\left(\pi|_{\boldsymbol{x}_{\backslash i}}, \mu|_{\boldsymbol{x}_{\backslash i}}, \boldsymbol{\theta}, 0.05 \cdot \frac{1}{T_1}\right)$ rejects, where $\boldsymbol{\theta} \sim \text{Unif}([0.05\varepsilon/c^*, \log(1/\eta)])$. If the total number of Coordinate Oracle calls needed to perform these calls to KL-TEST exceeds

$$T = \underbrace{100 \cdot T_1 \cdot c_{\text{KL-TEST}} \cdot \sqrt{|\Sigma|} \cdot \log(1/\eta) \cdot \log(20T_1) \cdot 10\log\left(\frac{\log(1/\eta)}{\varepsilon/c^*}\right)}_{\text{bounds expected number of Coordinate Oracle uses per call to KL-TEST}},$$

then reject. Here $c_{\text{KL-TEST}}$ is the constant hidden in the $O$-notation of Theorem 27.

2. Consider the distribution $\rho_\pi$ on $[k]$ defined by drawing $\boldsymbol{x} \sim \pi$ and outputting $a \in [k]$ with probability $\frac{\rho(a)\mu_a(\boldsymbol{x})}{\mu(\boldsymbol{x})}$. (Note the algorithm can simulate draws from $\rho_\pi$ using the General Oracle for $\pi$.) Reject if KL-TEST$(\rho_\pi, \rho, 0.5\varepsilon, 0.1)$ rejects. Otherwise accept.

---

We now prove that Algorithm 1 is a good identity-tester for the distribution $\mu$, deferring proofs of the auxiliary lemmas Lemma 33 and Lemma 34 to Appendix A.1.

**Theorem 29** (Theorem 5, restated). *Let* $\mu = \sum_{a=1}^{k} \rho(a)\mu_a$ *be* $\eta$-balanced, where $\rho^* = \min_a \rho(a)$ *and each* $\mu_a$ *satisfies* $c^*$-ATE. *Then Algorithm 1 uses*

$$\underbrace{O\left(\frac{c^*n}{\varepsilon} \cdot \log^2\left(\frac{c^*n}{\varepsilon}\right) \cdot \sqrt{|\Sigma|} \cdot \log(1/\eta) \cdot \log\log(1/\eta)\right)}_{\text{from Step 1}} + \underbrace{O\left(\frac{\sqrt{k} \cdot \log(1/\rho^*)}{\varepsilon}\right)}_{\text{from Step 2}}$$

*calls to the General Oracle and Coordinate Oracle and satisfies the following performance guarantees:*

1. *If* $\pi = \mu$ *then* COORDINATE-ORACLE-TEST$(\mu, \pi, \varepsilon)$ *rejects with probability at most* $0.4$.

2. *If* $D_{\text{KL}}(\pi \parallel \mu) \geq \varepsilon$ *then* COORDINATE-ORACLE-TEST$(\mu, \pi, \varepsilon)$ *rejects with probability at least* $0.6$.

*Proof.* Recalling that $T_1 = O\left(\frac{c^*n}{\varepsilon}\right)$, Step 1 contributes at most the following amount of calls to the oracles:

$$T = 1000 \cdot c_{\text{KL-TEST}} \cdot T_1 \cdot \sqrt{|\Sigma|} \cdot \log(1/\eta) \cdot \log(20T_1) \cdot \log\left(\frac{c^* \log(1/\eta)}{\varepsilon}\right).$$

$$= O\left(\frac{c^*n}{\varepsilon}\right) \cdot \log\left(\frac{c^*n}{\varepsilon}\right) \cdot \sqrt{|\Sigma|} \cdot \log(1/\eta) \cdot \log\left(\frac{c^* \log(1/\eta)}{\varepsilon}\right)$$

$$= O\left(\frac{c^*n}{\varepsilon} \cdot \log^2\left(\frac{c^*n}{\varepsilon}\right) \cdot \sqrt{|\Sigma|} \cdot \log(1/\eta) \cdot \log\log(1/\eta)\right).$$

Step 2 in Algorithm 1 uses the following amount of oracle calls:

$$O\left(\frac{\sqrt{k} \cdot \log(1/\rho^*)}{\varepsilon}\right).$$

Summing these gives the final number of oracle calls.

Now we bound the failure probabilities. Suppose that $\pi = \mu$ so that $D_{\mathrm{KL}}(\pi \parallel \mu) = 0$. Then the probability that Step 1 in Algorithm 1 rejects is at most 0.1 by Lemma 33. The probability that Step 2 in Algorithm 1 rejects is at most 0.1 by Lemma 34. Therefore, KL-TEST$(\mu, \pi, \varepsilon)$ accepts with probability at least 0.8.

Now suppose that $D_{\mathrm{KL}}(\pi \parallel \mu) = \mathbf{Ent}_\mu\left[\frac{\pi}{\mu}\right] \geq \varepsilon$. Using the chain rule (Lemma 18), we have

$$\mathbf{Ent}_\mu\left[\frac{\pi}{\mu}\right] = \mathop{\mathbf{E}}_{a \sim \rho}\left[\mathbf{Ent}_{\mu_a}\left[\frac{\pi}{\mu}\right]\right] + \mathbf{Ent}_{a \sim \rho}\left[\mathop{\mathbf{E}}_{\mu_a}\left[\frac{\pi}{\mu}\right]\right],$$

and we find that one of the two summands on the right-hand side is at least $0.5\varepsilon$. In the first case, we have $\mathbf{E}_{a \sim \rho}\left[\mathbf{Ent}_{\mu_a}\left[\frac{\pi}{\mu}\right]\right] \geq 0.5\varepsilon$ and then Lemma 33 shows that Step 1 rejects with probability at least 0.9. Otherwise if $\mathbf{Ent}_{a \sim \rho}\left[\mathbf{E}_{\mu_a}\left[\frac{\pi}{\mu}\right]\right] \geq 0.5\varepsilon$ then Lemma 34 shows that Step 2 rejects with probability at least 0.9. $\square$

## A.1. Proofs of Lemmas 33 and 34

In this section we prove Lemmas 33 and 34 which were the main tools needed in our application to identity-testing.

As in (Blanca et al., 2023), our sample complexity has a dependence on the balancedness of our visible distributions:

**Definition 30.** We say that a distribution $\mu$ on $\Sigma^n$ is $\eta$-*balanced* if for all $x \in \Sigma^n$ we have that the distribution $\mu|_{x_{\setminus i}}$ has minimum probability $\eta$.

We always regard the distribution $\mu$ on $\Sigma^n$ as a mixture of distributions $\mu = \sum_{a=1}^k \rho(a)\mu_a$, where each $\mu_a$ satisfies approximate tensorization of entropy with constant $c_a$. Moreover, we assume that $\mu$ is $\eta$-balanced. The following Fact 31 also shows that $\mu$ is fully supported whenever $\eta > 0$:

**Fact 31.** Let $\mu$ be an $\eta$-balanced distribution on $\Sigma^n$. Then for all $x \in \Sigma^n$, we have $\mu(x) \geq \eta^n$.

*Proof.* We induct on $n$. In the base case $n = 1$ and the conclusion is immediate.

For the inductive step let $\mu$ be an $\eta$-balanced distribution on $\Sigma^n$. For each $b \in \Sigma$, let $S_b = \{x \in \Sigma^n : x_n = b\}$. It must be the case that for all $b \in \Sigma$, we have $\mu(S_b) \geq \eta$, since otherwise by averaging there would be $y \in \Sigma^{n-1}$ such that $\mu|_{y_{\setminus n}}$ is not $\eta$-balanced. By the inductive hypothesis, since the distribution $\mu|_{S_b}$ is $\eta$-balanced, for any $x \in S_b$ we have $\mu|_{S_b}(x) \geq \eta^{n-1}$. Then $\mu(x) = \mu(S_b)\mu|_{S_b}(x) \geq \eta \cdot \eta^{n-1} = \eta^n$. $\square$

Throughout this section let $0 < \varepsilon < n \log\left(\frac{1}{\eta}\right)$, which is without loss of generality since if $\mu$ is $\eta$-balanced then by Fact 31 we have $\min_{x \in \Sigma^n} \mu(x) \geq \eta^n$. For any other distribution $\pi$, we have

$$D_{\mathrm{KL}}(\pi \parallel \mu) = \mathop{\mathbf{E}}_\pi\left[\log\left(\frac{\pi}{\mu}\right)\right] \leq \max_x \left(\log\left(\pi(x)\right)\mu(x)\right) \leq n \log\left(\frac{1}{\eta}\right).$$

Let $\pi$ be an arbitrary distribution on $\Sigma^n$, and recall that $\mu = \sum_{a=1}^k \rho(a)\mu_a$ is a mixture of $k$ distributions $\mu_1, \ldots, \mu_k$. Let $\rho^* = \min_a \rho(a)$.

### A.1.1. REJECTION BY LOCAL TESTING

**Lemma 32.** The probability that the number of calls to the Coordinate Oracle and General Oracle needed exceeds the limit stated in Step 1 is at most 0.01.

*Proof.* With certainty over the choice of $\boldsymbol{x}$ and $\boldsymbol{i}$, the distribution $\mu|_{\boldsymbol{x}_{\setminus i}}$ is has minimum probability at least $\eta$ by definition of $\eta$-balancedness. Then a call to KL-TEST$\left(\pi|_{\boldsymbol{x}_{\setminus i}}, \mu|_{\boldsymbol{x}_{\setminus i}}, \boldsymbol{\theta}, 0.05 \cdot \frac{1}{T_1}\right)$, by Theorem 27 requires

$$c_{\text{KL-TEST}} \cdot \frac{\sqrt{|\Sigma|} \cdot \log(1/\eta) \cdot \log(20T_1)}{\boldsymbol{\theta}}$$

calls to the Coordinate Oracle. To compute the total expected number of samples, we first compute

$$\mathbf{E}_{\theta}\left[\frac{1}{\theta}\right] = \frac{1}{\log(1/\eta) - 0.05\varepsilon/c^*} \int_{\frac{0.05\varepsilon}{c^*n}}^{\log(1/\eta)} \frac{1}{\theta} d\theta$$

$$= \frac{1}{\log(1/\eta) - 0.05\varepsilon/c^*} \ln\left(\frac{c^* \log(1/\eta)}{0.05\varepsilon}\right)$$

$$\leq \frac{10}{\log(1/\eta)} \log\left(\frac{c^* \log(1/\eta)}{\varepsilon}\right).$$

Here we use that $c^* \geq 1$, and that $\varepsilon \leq n \log\left(\frac{1}{\eta}\right)$ so that

$$\frac{0.05c^*\varepsilon}{n} \leq \frac{0.05n \log\left(\frac{1}{\eta}\right)}{n} = 0.05 \log\left(\frac{1}{\eta}\right).$$

Then, using that $\eta \leq \frac{1}{2}$, we bound this by

$$\mathbf{E}_{\theta}\left[\frac{1}{\theta}\right] \leq 10 \log\left(\frac{\log(1/\eta)}{c^*\varepsilon}\right).$$

Therefore, the expected number of samples, over all $T_1$ calls to KL-TEST is at most

$$T_1 \cdot 10 \cdot c_{\text{KL-TEST}} \cdot \sqrt{|\Sigma|} \cdot \log(1/\eta) \cdot \log(20T_1) \cdot \log\left(\frac{\log(1/\eta)}{c^*\varepsilon}\right).$$

Markov's inequality shows the result. $\qquad\square$

**Lemma 33.** Assume that $\mu$ is $\eta$-balanced. Then Algorithm 1 satisfies the following performance guarantees:

1. If $\pi = \mu$ then Step 1 in COORDINATE-ORACLE-TEST$(\mu, \pi, \varepsilon)$ rejects with probability at most $0.1$.

2. If $\mathbf{E}_{a \sim \rho}\left[\mathbf{Ent}_{x \sim \mu_a}\left[\frac{\pi(x)}{\mu(x)}\right]\right] \geq 0.5\varepsilon$ then Step 1 in COORDINATE-ORACLE-TEST$(\mu, \pi, \varepsilon)$ rejects with probability at least $0.9$.

*Proof.* Define the random variable $Y$ to be $\mathbf{Ent}_{z \sim \mu|_{x_{\setminus i}}}\left[\frac{\pi|_{x_{\setminus i}}(z)}{\mu|_{x_{\setminus i}}(z)}\right]$ for $x \sim \pi$ and $i \in [n]$ uniform.

Consider the version of Step 1 without the sample limit. If $\pi = \mu$ then by the guarantee of Theorem 27 shows that the rejection probability in each of the $T_1$ iterations using this unlimited tester in Step 1 is at most $0.05 \cdot \frac{1}{T_1}$. Since by Lemma 32 the limit on the number of Coordinate Oracle calls changes the behavior with probability at most $0.01$, the overall rejection probability is by a union bound at most $T_1 \cdot \frac{0.05}{T_1} + 0.01 \leq 0.1$.

Now suppose that $\mathbf{E}_{a \sim \rho}\left[\mathbf{Ent}_{x \sim \mu_a}\left[\frac{\pi(x)}{\mu(x)}\right]\right] \geq 0.5\varepsilon$. Then by Lemma 35 and Lemma 19 we have

$$\mathbf{E}\left[Y\right] = \frac{1}{n} \sum_{i \in [n]} \mathbf{E}_{x \sim \pi}\left[\mathbf{Ent}_{z \sim \mu|_{x_{\setminus i}}}\left[\frac{\pi|_{x_{\setminus i}}(z)}{\mu|_{x_{\setminus i}}(z)}\right]\right] = \frac{1}{n}\mathcal{L}_{\mu}\left[\frac{\pi}{\mu}\right] \geq \frac{0.5\varepsilon}{c^*n},$$

where $c^* = \max_a c_a$ is a constant such that all $\mu_a$ satisfy $c^*$-ATE.

Using that $Y \le \log(1/\eta)$ with certainty,

$$
\begin{aligned}
\mathbf{Pr}_{\theta,Y}\left[Y \ge \theta\right] &= \frac{1}{\log(1/\eta) - \frac{0.05\varepsilon}{c^*n}} \int_{\frac{0.05\varepsilon}{c^*n}}^{\log(1/\eta)} \mathbf{Pr}_Y\left[Y \ge \theta\right] d\theta \\
&\ge \frac{1}{\log(1/\eta)} \int_0^{\log(1/\eta)} \mathbf{Pr}_Y\left[Y \ge \theta\right] d\theta - \frac{1}{\log(1/\eta)} \int_0^{\frac{0.05\varepsilon/c^*}{n}} \mathbf{Pr}_Y\left[Y \ge \theta\right] d\theta \\
&= \mathbf{E}\left[Y\right] - \frac{1}{\log(1/\eta)} \int_0^{0.05\varepsilon/c^*} \mathbf{Pr}_Y\left[Y \ge \theta\right] d\theta \\
&\ge \mathbf{E}\left[Y\right] - \frac{0.05\varepsilon}{c^*n} \\
&\ge \frac{0.9}{n} \cdot \mathbf{E}\left[Y\right] \\
&\ge \frac{0.45\varepsilon}{c^*n}.
\end{aligned}
$$

Therefore, for $T_1$ draws of $\boldsymbol{x} \sim \pi$, $\boldsymbol{i} \sim [n]$ uniform, and $\boldsymbol{\theta}$, the probability that none satisfy $\boldsymbol{\theta} \le \mathbf{Ent}_{\boldsymbol{z} \sim \mu|_{\boldsymbol{x}\setminus i}}\left[\frac{\pi|_{\boldsymbol{x}\setminus i}(\boldsymbol{z})}{\mu|_{\boldsymbol{x}\setminus i}(\boldsymbol{z})}\right]$ is at most

$$
\left(1 - \frac{0.45\varepsilon}{c^*n}\right)^{T_1} \le 0.04
$$

for a correct choice of constant in the definition of $T_1$.

Let $\mathcal{A}$ be the event that this occurs so that $\mathbf{Pr}\left[\mathcal{A}\right] \le 0.01$. Then let $\mathcal{B}$ be the event that for some call $\text{KL-TEST}\left(\pi|_{\boldsymbol{x}\setminus i}, \mu|_{\boldsymbol{x}\setminus i}, \boldsymbol{\theta}, 0.05 \cdot \frac{1}{T_1}\right)$ for which $\boldsymbol{\theta} \le \mathbf{Ent}_{\boldsymbol{z} \sim \mu|_{\boldsymbol{x}\setminus i}}\left[\frac{\pi|_{\boldsymbol{x}\setminus i}(\boldsymbol{z})}{\mu|_{\boldsymbol{x}\setminus i}(\boldsymbol{z})}\right]$, the test accepted. The probability that a single call of this form failed is at most $0.05 \cdot \frac{1}{T_1}$, so a union bound gives $\mathbf{Pr}\left[\mathcal{B}\right] \le 0.05$.

By a union bound, the probability of accepting $\pi$ is then

$$
\mathbf{Pr}\left[\mathcal{A}\right] + \mathbf{Pr}\left[\mathcal{B}\right] \le 0.1.
$$

Thus, the performance guarantee is satisfied. $\qquad\square$

### A.1.2. REJECTION BY POSTERIOR WEIGHT ESTIMATION

**Lemma 34.** Algorithm 1 satisfies the following performance guarantees:

1. If $\pi = \mu$ then Step 2 in COORDINATE-ORACLE-TEST$(\mu, \pi, \varepsilon)$ rejects with probability at most $0.1$.

2. If $\mathbf{Ent}_{\boldsymbol{a} \sim \rho}\left[\mathbf{E}_{x \sim \mu_{\boldsymbol{a}}}\left[\frac{\pi(x)}{\mu(x)}\right]\right] \ge 0.5\varepsilon$ then Step 2 in COORDINATE-ORACLE-TEST$(\mu, \pi, \varepsilon)$ rejects with probability at least $0.9$.

*Proof.* By Fact 20 we have that

$$
\mathbf{Ent}_{\boldsymbol{a} \sim \rho}\left[\mathbf{E}_{x \sim \mu_{\boldsymbol{a}}}\left[\frac{\pi(x)}{\mu(x)}\right]\right] = \mathbf{Ent}_{\boldsymbol{a} \sim \rho}\left[\frac{\rho_\pi(\boldsymbol{a})}{\rho(\boldsymbol{a})}\right] = D_{\mathrm{KL}}(\rho_\pi \| \rho).
$$

Then the guarantee of Theorem 27 shows that if $\pi = \mu$ the step Step 2 rejects with probability at most $0.1$. Otherwise, if $\text{KL}(\rho_\pi, \rho) \ge 0.5\varepsilon$ then Step 2 in Algorithm 1 rejects with probability at least $0.9$. $\qquad\square$

**Lemma 35.** The following equality holds for all distribution $\pi$ and $\mu$ on $\Sigma^n$:

$$
\mathcal{L}_\pi\left[\frac{\pi}{\mu}\right] = \sum_{i \in [n]} \mathbf{E}_{\boldsymbol{x} \sim \pi}\left[\mathbf{Ent}_{\boldsymbol{z} \sim \mu|_{\boldsymbol{x}\setminus i}}\left[\frac{\pi|_{\boldsymbol{x}\setminus i}(\boldsymbol{z})}{\mu|_{\boldsymbol{x}\setminus i}(\boldsymbol{z})}\right]\right].
$$

*Proof.* We directly compute

$$\sum_{i\in[n]} \mathop{\mathbf{E}}_{\boldsymbol{x}\sim\pi}\left[\mathop{\mathbf{Ent}}_{\boldsymbol{z}\sim\mu|_{\boldsymbol{x}_{\backslash i}}}\left[\frac{\pi|_{\boldsymbol{x}_{\backslash i}}(\boldsymbol{z})}{\mu|_{\boldsymbol{x}_{\backslash i}}(\boldsymbol{z})}\right]\right] = \frac{1}{|\Sigma|}\sum_{i\in[n]}\sum_x \pi(x_{\backslash i})\mathop{\mathbf{Ent}}_{\boldsymbol{z}\sim\mu|_{x_{\backslash i}}}\left[\frac{\pi|_{x_{\backslash i}}(\boldsymbol{z})}{\mu|_{x_{\backslash i}}(\boldsymbol{z})}\right]$$

$$= \frac{1}{|\Sigma|}\sum_{i\in[n]}\sum_x \pi(x_{\backslash i})\cdot\frac{\mu(x_{\backslash i})}{\pi(x_{\backslash i})}\cdot\mathop{\mathbf{Ent}}_{\boldsymbol{y}\sim\mu|_{x_{\backslash i}}}\left[\frac{\pi(\boldsymbol{y})}{\mu(\boldsymbol{y})}\right]$$

$$= \frac{1}{|\Sigma|}\sum_{i\in[n]}\sum_x \mu(x_{\backslash i})\cdot\mathop{\mathbf{Ent}}_{\boldsymbol{y}\sim\mu|_{x_{\backslash i}}}\left[\frac{\pi(\boldsymbol{y})}{\mu(\boldsymbol{y})}\right]$$

$$= \mathcal{L}_\pi\left[\frac{\pi}{\mu}\right].$$

The second equality follows by 1-homogeneity of $\mathbf{Ent}[\cdot]$ (Fact 10). $\qquad\square$

**Remark 36.** Lemma 35 is the only place we require the $\Phi$-entropy we use to be $\Phi(u) = u\log u$, since this is the only place we need 1-homogeneity. We leave it open for future work whether one can obtain testers for different $\Phi$-entropies (i.e., other divergences besides KL-divergence) by bypassing the need for Lemma 35.

# B. Deferred Proofs

### B.1. Proof of Lemma 18

*Proof of Lemma 18.* We directly compute:

$$\mathop{\mathbf{Ent}}_{\boldsymbol{a}\sim\rho}^{\Phi}\left[\mathop{\mathbf{E}}_{\boldsymbol{x}\sim\mu_{\boldsymbol{a}}}[f]\right] + \mathop{\mathbf{E}}_{\boldsymbol{a}\sim\rho}\left[\mathop{\mathbf{Ent}}_{\boldsymbol{x}\sim\mu_{\boldsymbol{a}}}^{\Phi}[f]\right]$$

$$= \mathop{\mathbf{E}}_{\boldsymbol{a}\sim\rho}\left[\Phi\left(\mathop{\mathbf{E}}_{\boldsymbol{x}\sim\mu_{\boldsymbol{a}}}[f]\right)\right] - \Phi\left(\mathop{\mathbf{E}}_{\boldsymbol{a}\sim\rho}\mathop{\mathbf{E}}_{\boldsymbol{x}\sim\mu_{\boldsymbol{a}}}[f]\right) + \mathop{\mathbf{E}}_{\boldsymbol{a}\sim\rho}\left[\mathop{\mathbf{E}}_{\boldsymbol{x}\sim\mu_{\boldsymbol{a}}}[\Phi(f(\boldsymbol{x}))]\right] - \mathop{\mathbf{E}}_{\boldsymbol{a}\sim\rho}\left[\Phi\left(\mathop{\mathbf{E}}_{\boldsymbol{x}\sim\mu_{\boldsymbol{a}}}[f]\right)\right]$$

$$= \mathop{\mathbf{E}}_{\boldsymbol{a}\sim\rho}\left[\mathop{\mathbf{E}}_{\boldsymbol{x}\sim\mu_{\boldsymbol{a}}}[\Phi(f(\boldsymbol{x}))]\right] - \Phi\left(\mathop{\mathbf{E}}_{\boldsymbol{a}\sim\rho}\mathop{\mathbf{E}}_{\boldsymbol{x}\sim\mu_{\boldsymbol{a}}}[f]\right)$$

$$= \mathop{\mathbf{E}}_{\boldsymbol{x}\sim\mu}[\Phi(f(\boldsymbol{x}))] - \Phi\left(\mathop{\mathbf{E}}_{\boldsymbol{x}\sim\mu}[f]\right)$$

$$= \mathop{\mathbf{Ent}}_{\mu}^{\Phi}[f]. \qquad\square$$

### B.2. Proof of Lemma 19

*Proof of Lemma 19.* We compute

$$\mathcal{L}_\mu[f] = \sum_{i\in[n]}\mathop{\mathbf{E}}_{\boldsymbol{x}\sim\mu}\left[\mathop{\mathbf{Ent}}_{\boldsymbol{y}\sim\mu|_{\boldsymbol{x}_{\backslash i}}}[f(\boldsymbol{y})]\right] = \sum_{i\in[n]}\mathop{\mathbf{E}}_{\boldsymbol{a}\sim\rho,\boldsymbol{x}\sim\mu_{\boldsymbol{a}}}\left[\mathop{\mathbf{Ent}}_{\boldsymbol{y}\sim\mu|_{\boldsymbol{x}_{\backslash i}}}[f(\boldsymbol{y})]\right].$$

Since $\mu|_{\boldsymbol{x}_{\backslash i}} = \mathbf{E}_{\boldsymbol{a}'\sim\rho|_{\boldsymbol{x}_{\backslash i}}}[\mu_{\boldsymbol{a}'}|_{\boldsymbol{x}_{\backslash i}}]$, we have by the chain rule (Lemma 18) that the above is bounded below by

$$\sum_{i\in[n]}\mathop{\mathbf{E}}_{\boldsymbol{a}\sim\rho,\boldsymbol{x}\sim\mu_{\boldsymbol{a}}}\left[\mathop{\mathbf{E}}_{\boldsymbol{a}'\sim\rho|_{\boldsymbol{x}_{\backslash i}}}\left[\mathop{\mathbf{Ent}}_{\boldsymbol{y}\sim\mu_{\boldsymbol{a}'}|_{\boldsymbol{x}_{\backslash i}}}[f(\boldsymbol{y})]\right]\right] = \sum_{i\in[n]}\mathop{\mathbf{E}}_{\boldsymbol{a}\sim\rho,\boldsymbol{x}\sim\mu_{\boldsymbol{a}},\boldsymbol{a}'\sim\rho|_{\boldsymbol{x}_{\backslash i}}}\left[\mathop{\mathbf{Ent}}_{\boldsymbol{y}\sim\mu_{\boldsymbol{a}'}|_{\boldsymbol{x}_{\backslash i}}}[f(\boldsymbol{y})]\right]$$

$$= \sum_{i\in[n]}\mathop{\mathbf{E}}_{\boldsymbol{a}\sim\rho,\boldsymbol{x}\sim\mu_{\boldsymbol{a}}}\left[\mathop{\mathbf{Ent}}_{\boldsymbol{y}\sim\mu_{\boldsymbol{a}}|_{\boldsymbol{x}_{\backslash i}}}[f(\boldsymbol{y})]\right].$$

By applying ATE for each $\mu_{\boldsymbol{a}}$, we can lower-bound this by

$$\mathop{\mathbf{E}}_{\boldsymbol{a}\sim\rho}\left[\sum_{i\in[n]}\mathop{\mathbf{E}}_{\boldsymbol{x}\sim\mu_{\boldsymbol{a}}}\left[\mathop{\mathbf{Ent}}_{\boldsymbol{y}\sim\mu_{\boldsymbol{a}}|_{\boldsymbol{x}_{\backslash i}}}[f(\boldsymbol{y})]\right]\right] \geq \mathop{\mathbf{E}}_{\boldsymbol{a}\sim\rho}\left[\frac{1}{c^*}\cdot\mathop{\mathbf{Ent}}_{\boldsymbol{x}\sim\mu_{\boldsymbol{a}}}[f(\boldsymbol{x})]\right]. \qquad\square$$

## B.3. Proof of Lemma 21

*Proof of Lemma 21.* Let $\pi$ be any initial distribution and let $f = \pi/\mu$ be the density function of $\pi$ with respect to $\mu$. The impetus behind the switch to continuous-time $P_t$ rather than discrete-time $P$ is that we immediately get the following continuous characterization of $\Phi$-divergence contraction:

**Lemma 37** ((Chafaï, 2004), Proposition 1). Let $f_t = P_t f$. Then,

$$\frac{d}{dt} D_\Phi(P_t \pi \parallel \mu) = -\mathcal{E}_P(f_t, \Phi'(f_t)).$$

This turns out to be the only piece of the proof missing towards a generalization of Theorem 4.5 from (Huang et al., 2024). All that is left is to establish a *weak $\Phi$-Sobolev inequality* for mixtures.

**Definition 38.** A distribution $\mu$ on $\Sigma^n$ satisfies a *weak $\Phi$-Sobolev inequality* with constant $c^*$ and error $g : \mathbb{R}_{\geq 0}^{\Sigma^n} \to \mathbb{R}_{\geq 0}$ if for all $f : \Sigma^n \to \mathbb{R}_{\geq 0}$,

$$\mathbf{Ent}_\mu^\Phi[f] \leq c^* \cdot \mathcal{E}_P(f, \Phi'(f)) + g(f).$$

Let $f : \Sigma^n \to \mathbb{R}_{\geq 0}$ and observe

$$
\begin{aligned}
\mathbf{Ent}_\mu^\Phi[f] &= \underset{\boldsymbol{a} \sim \rho}{\mathbf{E}} \left[ \underset{\boldsymbol{x} \sim \mu_{\boldsymbol{a}}}{\mathbf{Ent}^\Phi}[f] \right] + \underset{\boldsymbol{a} \sim \rho}{\mathbf{Ent}^\Phi} \left[ \underset{\boldsymbol{x} \sim \mu_{\boldsymbol{a}}}{\mathbf{E}} [f] \right] \\
&\leq c^* \cdot \underset{\boldsymbol{a} \sim \rho}{\mathbf{E}} [\mathcal{E}_{P_{\boldsymbol{a}}}(f, \Phi'(f))] + \underset{\boldsymbol{a} \sim \rho}{\mathbf{Ent}^\Phi} \left[ \underset{\boldsymbol{x} \sim \mu_{\boldsymbol{a}}}{\mathbf{E}} [f] \right] \\
&\leq c^* \cdot \mathcal{E}_P(f, \Phi'(f)) + \underset{\boldsymbol{a} \sim \rho}{\mathbf{Ent}^\Phi} \left[ \underset{\boldsymbol{x} \sim \mu_{\boldsymbol{a}}}{\mathbf{E}} [f] \right].
\end{aligned}
$$

The first line is the chain rule (Lemma 18). The second line applies the component-wise $\Phi$-Sobolev inequality. The third line invokes the concavity of the Dirichlet form for Glauber dynamics. That is, we use that

$$
\begin{aligned}
\mathcal{E}_P(f, \Phi'(f)) &= \underset{\boldsymbol{x}}{\mathbf{E}} \underset{\boldsymbol{y} \sim P \boldsymbol{x}}{\mathbf{E}} [(f(\boldsymbol{x}) - f(\boldsymbol{y}))(\Phi'(f(\boldsymbol{x})) - \Phi'(f(\boldsymbol{y})))] \\
&= \frac{1}{n} \sum_{x \sim y} \frac{\mu(x)\mu(y)}{\mu(x) + \mu(y)} (f(x) - f(y))(\Phi'(f(x)) - \Phi'(f(y))) \\
&\geq \frac{1}{n} \sum_{x \sim y} \underset{\boldsymbol{a}}{\mathbf{E}} \left[ \frac{\mu_{\boldsymbol{a}}(x)\mu_{\boldsymbol{a}}(y)}{\mu_{\boldsymbol{a}}(x) + \mu_{\boldsymbol{a}}(y)} \right] (f(x) - f(y))(\Phi'(f(x)) - \Phi'(f(y))).
\end{aligned}
$$

Here the inequality follows from concavity of the map $(a, b) \mapsto \frac{ab}{a+b}$ and the fact that $\Phi'$ is increasing in $f(\cdot)$, so the summands are all positive.

With this weak $\Phi$-Sobolev inequality and Lemma 37 the result follows by the exact proof of Theorem 4.5 from (Huang et al., 2024), replacing KL-divergence with $\Phi$-divergence and $\mathcal{E}_P(f_t, \log f_t)$ with $\mathcal{E}_P(f_t, \Phi'(f_t))$. □

## B.4. Proof of Theorem 27

*Proof of Theorem 27.* Irrespective of $q$ having minimum probability $\eta$, Theorem 1 of (Daskalakis et al., 2018)[3] gives an algorithm we'll call $H^2$-TEST$(p, q, \varepsilon, \delta)$ that — given $q, \varepsilon, \delta$ and samples from an unknown $p$ on $D$ — has sample complexity $O\left( \frac{\sqrt{d} \cdot \log(1/\delta)}{\varepsilon} \right)$ and the following guarantee:

1. If $\chi^2(p \| q) \leq 0.5\varepsilon$ (e.g., if $p = q$), $H^2$-TEST$(p, q, \varepsilon, \delta)$ accepts with probability at least $1 - \delta$.

2. If $H^2(p \| q) \geq \varepsilon$, $H^2$-TEST$(p, q, \varepsilon, \delta)$ accepts with probability at most $\delta$.

---

[3]See also (Acharya et al., 2015; Bădescu et al., 2019).

We also have the following inequality relating Hellinger distance and KL-divergence:

$$D_{\mathrm{H}^2}(p \parallel q) \geq \frac{1}{\log(e^2/\eta)} \cdot D_{\mathrm{KL}}(p \parallel q).$$

(With a slightly different constant factor, this inequality appears in, e.g., (Birgé & Massart, 1998). See Prop. 2.12 (Flammia & O'Donnell, 2024) for the version above.) It follows that we can simply run $H^2$-TEST$(\varepsilon / \log(e^2/\eta), p, q, \delta)$ to obtain the desired KL-TEST. $\qquad\square$

