# OpenReview forum: "Sampling and Identity-Testing Without Approximate Tensorization of Entropy"
_ICML.cc/2026/Conference — ICML 2026 regular_

### Official Review · Reviewer_zK2b · 2026-03-06

**Soundness:** 3
**Presentation:** 3
**Significance:** 3
**Originality:** 3
**Overall Recommendation:** 5
**Confidence:** 3

**Summary:**

This paper is concerned with sampling from and testing the identity of a mixture of distributions, each of which satisfy approximate tensorization of entropy (ATE). Prior results have shown that these tasks, when considered for a single distribution satisfying ATE, can be efficiently solved. However, a mixture of distributions that invidiually satisfy ATE does not necessarily satisfy ATE, and prior guarantees do not port over. For a target unknown distribution that is a mixture of distributions individually satisfying ATE, the first main result in the paper (Theorem 4) shows that Glauber dynamics, seeded with the empirical distribution over a sufficiently large sample, mixes to a distribution that is close to the target distribution in KL divergence. The second main result (Theorem 5) gives an identity testing algorithm, which operates in a so-called coordinate-conditional sampling access model, and correctly determines from samples, whether the target distribution is equal to a known reference distribution, as opposed to being far from it in a given distance measure. The latter result resolves an open question posed in earlier work by Blanca et al. 2023. Together, these results extend prior results known for unimodal distributions satisfying ATE to mixtures of distributions that satisfy ATE.

**Compliance With Llm Reviewing Policy:**

Affirmed.

**Final Justification:**

The author rebuttal suitably addresses my questions, and I maintain my positive evaluation of this paper

**Key Questions For Authors:**

1) Could the authors briefly summarize the main technical innovations of the paper? Beyond employing and assimilating tools from prior literature, what are the main challenges that the authors had to overcome in their analyses?

2) Could the authors comment on whether the derived results extend to mixtures of distributions over a continous space? If not, what are the main challenges?

3) Given that the algorithm for sampling is essentially Glauber dynamics initialized at the empirical distribution, which seems within the scope of implementation, did the authors try to experiment on synthetic data generated from a mixture model?

**Limitations:**

yes

**Strengths And Weaknesses:**

### Strengths:

The paper is well-written, and has a clear contribution. It resolves an open problem from prior work, and also extends results for unimodal distributions to multi-modal distributions. The paper does a good job of placing the contributions of the paper (in terms of results) within the context of known literature.

### Weaknesses:

The paper would benefit from having a summarizing overview of techniques after the introduction. As written, it is hard for a reader to realize what new techniques were required to establish the results in the paper. Furthermore, the entire technical section for identity testing is in the Appendix. Given that the last page before references is essentially fully available to include more content from the Appendix, I would encourage the authors to include a sketch of the identity testing algorithm, as well as a proof overview of Theorem 5, in the main paper.

---

> ### Author Rebuttal · Authors · 2026-03-26
>
> We thank the reviewer for the comments and questions.
>
> We are happy to restructure the technical overview to better highlight our novel contributions.
> We view our primary contribution as a clean and concise formulation of the chain rule of entropy as a tool to study divergences between mixtures of well-behaved distributions. While the decomposition of variance via the law of total variance is present in previous work on related problems, our two conceptual contributions are the following:
>
> 1. The formulation of the ``inter-component" entropy as a divergence between posterior distributions was not present in previous works. This perspective was key to our results, as it allowed us to formulate dealing with the inter-component entropy as a testing/learning problem on the universe of mixture components.
>
> 2. We use that KL-divergence is more tightly related to concentration of empirical distributions than $\chi^2$ divergence is, and that this leads to the optimal sample complexity for our mixing result.
>
> We plan to include more of an overview of these techniques in the final version per your advice. We will also include a technical overview of the identity-testing section per yours and other reviewers' suggestion.
>
> Regarding extensions to distributions over continuous distributions, we did not explicitly work this out in the paper but our framework generalizes to these continuous domains as well as explored by Huang et al.~2024.
>
> We agree that the sampling task should be relatively easy to implement on, say, a mixture of two biased hypercubes, but we did not implement our results practically as we think the problem is mainly of theoretical interest.

---

> > ### Author Rebuttal · Reviewer_zK2b · 2026-04-03
> >
> > Thanks for the response. Given your comment about the results extending to continuous domains, I would definitely discuss this slightly more precisely in the updated version.
> >
> > I will maintain my positive review

---

### Official Review · Reviewer_rUnL · 2026-03-10

**Soundness:** 3
**Presentation:** 2
**Significance:** 3
**Originality:** 3
**Overall Recommendation:** 4
**Confidence:** 1

**Summary:**

This paper studies two fundamental algorithmic tasks, approximate sampling and identity testing, for mixture distributions whose components individually satisfy approximate tensorization of entropy (ATE). The central observation motivating the work is that while ATE greatly facilitates both tasks in the unimodal setting, mixtures of ATE distributions do not in general inherit ATE, leaving a theoretical gap for multimodal distributions.

The paper addresses this gap with two main contributions. First, it shows that Glauber dynamics initialized from an empirical data-based distribution mixes efficiently for mixtures satisfying modified log-Sobolev inequalities (MLSI), with sample complexity and mixing time given in Theorem 4, generalizing prior results of Koehler et al. (2024) and Huang et al. (2024) which worked under the weaker Poincaré inequality. Second, the paper answers an open question of Blanca et al. (2023) by providing an efficient identity tester for mixtures of ATE distributions in the coordinate-conditional sampling model, with oracle complexity given in Theorem 5.

The technical backbone of both results is the chain rule for entropy (Lemma 18), which decomposes divergence into intra-component and inter-component terms. The intra-component term is handled by adapting the existing ATE-based machinery, while the inter-component term is controlled via a moment generating function bound on the KL-divergence of an empirical posterior distribution (Lemma 24), drawing on a result of Agrawal (2020).

**Compliance With Llm Reviewing Policy:**

Affirmed.

**Final Justification:**

For the presentation of Theorem 5, the authors promise to revise the main draft to include a fuller proof sketch of Theorem~5 in the main text, which may better present the methodology. For the dependence on $k$ and the dependence on the minimum mixture weight. Other reviewers also raise similar questions, and I also checked authors' rebuttal to other reviewers on these similar concerns. I think my concern is addressed.

After consideration, I will keep my positive assessment.

**Key Questions For Authors:**

1. The $ O(\sqrt{k} \cdot \log(1/\rho^* )/ \epsilon) $ term in Theorem 5 comes from applying KL-TEST to the posterior distribution, which has domain size $k$. Is the dependence tight here?

2. Remark 6 notes that efficient computational complexity follows from efficiently implementable Glauber dynamics steps for $\mu$. Can the authors say more concretely about which classes of distributions satisfy this condition in practice?

**Limitations:**

The paper would benefit from a clearer presentation of the identity-testing algorithm in the main body, and a discussion of the tightness of the $\sqrt{k}$ dependence in the testing result.

**Strengths And Weaknesses:**

**Strength**

1. The intra/inter-component decomposition via the entropy chain rule is clean and unifying. It provides a transparent framework that governs both the sampling and testing results, and the paper communicates this structure well.

2. The paper achieves sample complexity that is optimal in the sense that it matches known lower bounds in special cases (e.g., mixtures of isolated point masses), which is a quantitative improvement over Huang et al. (2024).

3. Upgrading from Poincaré inequalities to MLSIs (which follow from ATE) is a genuine improvement: it yields optimal mixing times in the product distribution case, whereas prior work suffered from suboptimal mixing time. The authors are transparent about exactly where and why this improvement occurs.

**Weakness**

1. Theorem 5 is stated in the main body but its proof is deferred almost entirely to the appendix. Importantly, the algorithm (Algorithm 1) also only appears in the appendix. This makes Section 1.2 difficult to evaluate from the main text alone, and the open question resolution, which is advertised as a primary contribution, is somewhat undersupported in the main body.

2. Theorem 5's sample complexity includes a term that containing $\sqrt{k}$ and $\log(1 / \rho^* )$ from the inter-component testing step (Step 2 of Algorithm 1). When $k$ is large or $\rho^*$ is small (i.e., the distribution is highly imbalanced), this term can dominate, which may limits the practicality of the algorithm. The author are encouraged to discuss whether this dependence is necessary or whether it can be improved.

---

> ### Author Rebuttal · Authors · 2026-03-26
>
> On the identity-testing proof being deferred to the appendix:
> This is a fair suggestion, and we will revise the main draft to include a fuller proof sketch of Theorem~5 in the main text.
>
> Regarding the dependence on $k$ and the minimum mixture weight in the sample complexity for the testing result: it is not clear to us that a dependence on $k$ is necessary. If we remove the $\eta$-balanced condition, then there exist mixtures of $k$ ATE distributions that require $\sqrt{k}$ samples to test, and a dependence on the minimum mixture weight is unavoidable. Take, for example, $k$ point masses in the hypercube that are not adjacent to each other. Then the coordinate oracle is useless, and the problem reduces to identity-testing of a distribution on $k$ universe elements. However, the $\eta$-balanced condition disallows this example, and we are unaware of a lower bound of this form with $\eta$-balanced mixture components.
>
> On the implementability of Glauber dynamics mentioned in the discussion of the testing result: normally, when one implements Glauber dynamics with respect to a distribution on the hypercube, one has access to a full description of the distribution. For example, it may be the case that the distribution is a Gibbs distribution with respect to a local Hamiltonian $H$, and the implementer of Glauber dynamics has  a description of $H$. In our setting, we imagine an evolution according to the Glauber dynamics of an unknown distribution, and we imagine that perhaps Nature is implementing this dynamics for us. Indeed, this is a common dynamic model in statistical physics.

---

> > ### Author Rebuttal · Reviewer_rUnL · 2026-04-03
> >
> > I thank the authors for the clarifications.
> >
> > For the presentation of Theorem 5, the authors promise to revise the main draft to include a fuller proof sketch of Theorem~5 in the main text, which may better present the methodology.
> >
> > For the dependence on $k$ and the dependence on the minimum mixture weight. Other reviewers also raise similar questions, and I also checked authors' rebuttal to other reviewers on these similar concerns. I think my concern is addressed.
> >
> > After consideration, I will keep my assessment.

---

### Official Review · Reviewer_jQGn · 2026-03-12

**Soundness:** 3
**Presentation:** 3
**Significance:** 4
**Originality:** 3
**Overall Recommendation:** 4
**Confidence:** 3

**Summary:**

The paper gives two results for mixtures of ATE distributions on $\Sigma^n$: fast mixing of Glauber dynamics from a data-based initialization with optimal sample complexity O($k/ε$ + $\log(1/δ)/ε$), and an efficient identity-tester under coordinate-conditional access that resolves an open question from Blanca et al. (2023). Both use a chain rule decomposition to separate inter-component from intra-component entropy.

**Compliance With Llm Reviewing Policy:**

Affirmed.

**Final Justification:**

I have raised the score following the rebuttal positively.

**Key Questions For Authors:**

1. Can the testing algorithm work without knowing the mixture decomposition? How sensitive is it to misspecification of k?

2. Does the Φ-Sobolev generalization yield anything for continuous MCMC (Langevin, log-concave mixtures)?

3. Is the sample complexity tight in k? Is the √k in Theorem 5 line (3) necessary?

4. Concrete examples beyond high-temperature Ising where mixture-of-ATE structure arises and prior results fail?

**Limitations:**

Yes

**Strengths And Weaknesses:**

# Strength
The chain rule (Lemma 18) ties the two results together nicely — both reduce to the same structural split, giving the paper good coherence. The best technical move is Lemma 24, where a convexity argument applied to Agrawal's m.g.f. bound handles overlapping components and kills the $\rho^*$ dependence from Huang et al. (2024). That's clean and the right idea. Resolving the Blanca et al. open question is a real contribution the testing community will care about.

# Weaknesss

Novelty is my main concern. The sampling proof follows Huang et al.'s paradigm closely, the $\Phi-$Sobolev generalization (Lemma 21) reproves their Theorem 4.5 with a different divergence plugged in, and the testing algorithm adds one natural step (posterior weight estimation) to the existing Blanca et al. algorithm. The KL-tester (Theorem 27) combines two known results. Each piece works; none is a conceptual surprise. The contribution is in the assembly.

The testing result, the more distinctive one, gets shortchanged. Its proof is entirely in the appendix while the sampling result gets full main-body treatment. Both results also require knowing the full mixture decomposition $(k, \rho(a), \textnormal{each } \mu_a)$, which is a strong assumption the paper doesn't interrogate. The generative modeling motivation in the introduction points toward continuous settings that the paper never seems to addresses. Does the $\Phi-$Sobolev upgrade help for Langevin on log-concave mixtures? A comparison table against Koehler et al. and Huang et al. would also help Section 1.3.


## Typos and Minor Issues

- Remark 9 (p. 5): "defiend" → "defined."
- Section 1.4 (p. 4): "distributon" → "distribution."
- Algorithm 1 is named `PRODUCT-SET-KL-TEST` but the proof of Theorem 29 calls it `COORDINATE-ORACLE-TEST` with swapped argument order. Lemma 33's proof uses yet another name (`KL-TEST`). Pick one.
- Theorem 27 proof (p. 18): broken citation renders as `(?)Prop. 2.12]flammia2024quantum` — malformed LaTeX.
- Blanca et al. (2022) reference: garbled author name `vStefankovivc` (should be Štefankovič).

---

> ### Author Rebuttal · Authors · 2026-03-26
>
> We thank the reviewer for the comments and questions. We will correct the mistakes the reviewer found in our manuscript.
>
> Can the testing algorithm work without knowing the mixture decomposition? How sensitive is it to misspecification of $k$? Our testing algorithm is mostly a information-theoretic result.
> Suppose one has a full specification of the target distribution $\mu$ that is a mixture of $k$ ATE distributions, but you do not know the exact decomposition. Then in principle one can run a brute force algorithm to compute a decomposition of $\mu$ into ATE mixture components and then run our algorithm with this knowledge. However, this is computationally inefficient; if one has a description of the mixture weights and components then our algorithm can be made efficient. We do not deal with the computational issues of learning such a description.
>
> Does the $\Phi$-Sobolev generalization yield anything for continuous MCMC (Langevin, log-concave mixtures)? We did not explicitly work this out in the paper, but our framework generalizes to these continuous domains as well as explored in (Huang et al.).
>
> We think the idea of a comparison table is a great idea and will add it for the final version.
>
> Regarding the dependence on $k$ in the sample complexity for the mixing result: this dependence is necessary because in the case where the mixture components are completely separated in the hypercube, we essentially must draw samples whose empirical distribution on mixture components is close to the true distribution. Thus, we must have enough samples to learn the true distribution, and there is a lower bound for this task matching our sample complexity upper bound. See, for example https://arxiv.org/pdf/2002.11457. We will make this more clear in the final version of the paper.
>
> Regarding the dependence on $k$ in the sample complexity for the testing result: it is not clear to us that a dependence on $k$ is necessary. If we remove the $\eta$-balanced condition, then there exist mixtures of $k$ ATE distributions that require $\sqrt{k}$ samples to test. Take, for example, $k$ point masses in the hypercube that are not adjacent to each other. Then the coordinate oracle is useless, and the problem reduces to identity-testing of a distribution on $k$ universe elements. However, the $\eta$-balanced condition disallows this example, and we are unaware of a lower bound of this form with $\eta$-balanced mixture components.
>
> Finally, here is a concrete example of a situation in which mixture-of-ATE structure arises and prior results fail: a mixture of $k$ product distributions in the hypercube. Suppose $k=2$ and one distribution is a product distribution with i.i.d. $0.99$-biased bits and the other is a product distribution with i.i.d. $0.01$-biased bits. Then Glauber dynamics cannot mix fast from a worst-case initialization.

---

> > ### Author Rebuttal · Reviewer_jQGn · 2026-04-03
> >
> > I thank the authors for the detailed response. Most of my concerns from the questions section have been adequately addressed. I raise my score.

---

### Official Review · Reviewer_NeYG · 2026-03-13

**Soundness:** 4
**Presentation:** 3
**Significance:** 3
**Originality:** 4
**Overall Recommendation:** 5
**Confidence:** 3

**Summary:**

This work discusses the problem of sampling and testing from mixture models, whose components satisfy a functional inequality known as approximation tensorization of entropy. Despite the overall mixture lacking theoretical guarantees for the fast mixing of the underlying dynamics, the authors demonstrate the given a data-driven initialization  leads to efficient sampling and identity testing.

**Compliance With Llm Reviewing Policy:**

Affirmed.

**Final Justification:**

The rebuttal of the authors addressed my concerns and I keep my positive evaluation of the work. This paper is written in a polished manner, and fills an important gap within the literature. These things in mind, I recommend for acceptance.

**Key Questions For Authors:**

>- Is a sample complexity dependent on $k$ necessary? What if the mixture weights associated to a given mixture are exponentially small?

**Limitations:**

yes

**Strengths And Weaknesses:**

**Strengths**
>- The paper is well written and structured; the main results and proof steps behind them are presented in an easy to understand fashion.
>- In recent years, many sampling problems which are provably slow, have been shown to admit measure decompositions with favourable theoretical properties. This work is an important step towards understanding the relationship between local and global entropy, and beyond-worst case sampling from discrete measures.
>- In particular, the use of the modified log-Sobolev inequalities in the work makes the result sharp.

**Weaknesses**
>- The main draft could use an explanation of the proof of the identity-testing problem, in lieu of it being deferred to the appendix.
>- It would be nice to see an application of Theorem 4 to a model of interest that can be represented as a mixture? Does this result improve our ability to sample in a more concrete way?

---

> ### Author Rebuttal · Authors · 2026-03-26
>
> We thank the reviewer for the careful reading and for the positive assessment of the paper's technical quality, originality, and significance.
>
> On the identity-testing proof being deferred to the appendix:
> this is a fair suggestion, and we will revise the main draft to include a fuller proof sketch of Theorem~5 in the main text.
>
> Regarding applications of Theorem 4, previous works (namely Koehler et al.) explored a few applications: mixing of the Langevin dynamics with a suitable score function and learning for new classes of Ising models using well-studied measure decompositions. We believe that our results should be relevant in these settings as well, but we did not pursue them as our main technical contribution was to show that the theoretical framework of beyond unimodal sampling and testing works for ATE rather than just Poincaré inequalities, allowing applications like these as a downstream effect. There are also smaller simple and immediate cases like the mixture of biased hypercubes for which are results immediately yield faster mixing, but these are perhaps more of theoretical than practical interest.
>
> Regarding the dependence on $k$ in the sample complexity for the mixing result: this dependence is necessary because in the case where the mixture components are completely separated in the hypercube, we essentially must draw samples whose empirical distribution on mixture components is close to the true distribution. Thus, we must have enough samples to learn the true distribution, and there is a lower bound for this task matching our sample complexity upper bound. See, for example. https://arxiv.org/pdf/2002.11457. We will make this more clear in the final version of the paper.
>
> On tiny mixture weights: we do not have any dependence on the minimum mixture weight in our sample complexity. If some of your weights are exponentially small then you can indeed ignore that component of the mixture and the dependence on $k$ disappears. We did not highlight this in particular as our main goal was to handle the hard case, when the mixture weights are roughly even, at least up to arbitrarily large polynomial factors, for which our Theorem 4 achieves optimal sample complexity.

---

> > ### Author Rebuttal · Reviewer_NeYG · 2026-04-02
> >
> > This paper fills an important gap in the literature, and the rebuttal answered remaining questions I had about the work, and as such I maintain my positive score.

---

### Decision · Program_Chairs · 2026-04-30

**Decision:**

Accept (regular)

**Comment:**

This paper contributes a fundamental advance to the theory of sampling and testing with mixture models. As two reviewers put it, the paper "resolves an open problem from prior work" and "fills an important gap in the literature". While previous work highlights the importance of distributions satisfying a condition called the "approximate tensorization of entropy" (ATE) and establishes theoretical guarantees regarding sampling from and testing with them, the proposed paper extends these results to mixtures of such distributions, which may not satisfy ATE marginally. While some reviewers were initially concerned that the theory, while novel, did not require major conceptual leaps to develop, the authors were able to assuage those concerns during the discussion period. Ultimately all reviewers appreciated the significance of the paper's results and found the theory itself to be rigorous, elegant, and novel enough for publication.